# Towards Better Branching Policies: Leveraging the Sequential Nature of Branch-and-Bound Tree

## Abstract

The branch-and-bound (B&B) method is a dominant exact algorithm for solving Mixed-Integer Linear Programming problems (MILPs). While recent deep learning approaches have shown promise in learning branching policies using instance-independent features, they often struggle to capture the sequential decision-making nature of B&B, particularly over long horizons with complex inter-step dependencies and intra-step variable interactions. To address these challenges, we propose Mamba-Branching, a novel learning-based branching policy that leverages the Mamba architecture for efficient long-sequence modeling, enabling effective capture of temporal dynamics across B&B steps. Additionally, we introduce a contrastive learning strategy to pre-train discriminative embeddings for candidate branching variables, significantly enhancing Mamba's performance. Experimental results demonstrate that Mamba-Branching outperforms all previous neural branching policies on real-world MILP instances and achieves superior computational efficiency compared to the advanced open-source solver SCIP. The source code can be accessed via an anonymized repository at https://anonymous.4open.science/r/Mamba-Branching-B4B4/.

## 1   Introduction

Mixed Integer Linear Programming problems (MILPs) constitute a class of computationally challenging NP-hard problems with widespread applications across diverse domains, including scheduling [7], planning [37], and transportation [3]. The branch-and-bound (B&B) method [28] represents the predominant solution methodology for MILPs in practice. This approach begins with the relaxation of the original problem and iteratively branches on variables that violate integer constraints. By maintaining global upper and lower bounds, the method progressively converges toward an optimal solution. Many high-performance MILP solvers such as SCIP [6] and Gurobi [17] employ the B&B framework as their core solution architecture.

Within the B&B framework, the selection of branching variables plays a critical role in determining computational efficiency. To this end, learning-based branching methods have been proposed [12, 15, 16, 40]: by constructing a bipartite graph that incorporates instance features and intra-tree dynamic features, and utilizing graph convolutional networks (GCNN) [12] for state encoding. Nevertheless, reliance on instance-specific features restricts their generalization to heterogeneous MILP instances. To enable cross-instance adaptability, recent approaches have focused on parameterizing the B&B tree to construct a shared feature space independent of specific problem data. For example, Zarpellon et al. [48] develop a parameterized B&B tree framework to create a shared feature space, decoupling branching decisions from instance-specific features. Further advancing this approach, T-BranT [31] evaluates the mutual connections between candidate variables by the self-attention mechanism and employs Graph Attention Networks to encode the empirical branching history in the search tree.

However, existing works universally overlook the sequential nature inherent in B&B tree expansion. In this paper, our key insight lies in that the "branching path" from the root node to the optimal solution node essentially constitutes a serialization process. This "branching path", which encompasses the parameterized tree states and corresponding branching variables from each preceding step, significantly influences the current branching decision. While T-BranT incorporates historical data, it models the tree from an unordered graph perspective, failing to explicitly capture this essential sequential nature. Effectively modeling this sequential nature presents two key challenges: (1) Design of long sequence modeling architectures. The sequence model must simultaneously capture inter-step dependencies and intra-step candidate variable relationships. Given that each state comprises multiple candidate variables, the length of the sequence input will increase exponentially with the number of branching steps. Therefore, it is essential to develop specialized architectures that can accommodate ultra-long sequences. (2) Construction of discriminative feature embeddings. An embedding layer needs to be designed to map the features of candidate variables into a high-dimensional vector space with high discriminative power. This will enable the sequence model to effectively discern the dynamic evolution patterns of different variables.

To address these challenges, we propose Mamba-Branching. Mamba [14, 8] is a novel network architecture characterized by its computational complexity that scales linearly with sequence length. This represents a significant improvement over the quadratic complexity associated with Transformers [43], making Mamba particularly well-suited for addressing challenge (1). Meanwhile, inspired by CLIP [38], we employ contrastive learning to train the embedding layer prior to the overall imitation learning process, effectively tackling challenge (2). Experimental results demonstrate that Mamba-Branching outperforms all neural branching baselines across all real-world instances and achieves superior solving efficiency over the advanced open-source solver SCIP's default branching rule on challenging instances.

## 2   Related Work

Learning-based approaches for accelerating MILP solving can be mainly divided into two paradigms [5, 39]: replacing heuristic rules with neural networks within exact solution frameworks and employing neural networks as primal heuristics. Research under the first paradigm includes addressing branch variable selection [12, 15, 16, 40, 25, 26] and node selection [19, 27, 49] problems within the B&B framework, as well as tackling cut selection issues in cutting-plane algorithms [42, 22, 45]. These methods solely employ neural networks to replace heuristic rules within exact solution frameworks, without compromising solution exactness. The second paradigm aims to efficiently produce high-quality feasible solutions—rather than exact solutions—to tighten the primal bound early in the process. A high-quality primal bound enables the B&B to eliminate a significant number of non-promising nodes at an early stage through its pruning process. This typically involves two key aspects: solution prediction [9, 36, 24, 18, 21, 33] and neighborhood selection [46, 41, 20, 47]. The solution prediction approach typically employs neural networks to predict optimal solutions, then uses these predictions to guide the search process. Neighborhood selection starts from a feasible solution and fixes a subset of integer variables while optimizing the remainder, with neural networks selecting which variables to fix.

Our work focuses on the generalization of neural branching variable selection policies, particularly their ability to handle heterogeneous MILPs different from training instances. These approaches can be mainly divided into two categories: parameterizing the B&B tree and diversifying training instances. The first category aims to learn branching policies within a shared feature space across different MILP instances. TreeGate [48] processes instance-independent features through a specialized neural architecture designed for branching decisions. Building on this, T-BranT [31] retains historical data, modeling it as a graph structure processed by Graph Attention Networks for current decision-making. The second category focuses on generating diverse instances and incorporating them into the training of branching policies to enhance their generalization. AdaSolver [32] introduces adversarial instance augmentation, which generates more diverse instances in directions that hinder policy training. Meanwhile, MILP-Evolve [30] proposes a novel LLM-based evolutionary framework capable of generating a large set of diverse MILP classes with an unlimited number of instances. Specifically, our method falls into the first category. Using instance-independent features as input, we also incorporate the sequential nature of the B&B tree into the decision-making process.

## 3 Preliminaries

### 3.1 B&B Algorithm and Branching Rules

**B&B Algorithm.** The standard form of MILPs is: $\arg\min_{\mathbf{x}}\left\{\mathbf{c}^{\top}\mathbf{x}\mid\mathbf{A}\mathbf{x}\leq\mathbf{b},\mathbf{x}\in\mathbb{Z}^{p}\times\mathbb{R}^{n-p}\right\}$, where the vector $\mathbf{x}$ represents $n$ variables to be optimized, with $p$ being the number of integer variables. $\mathbf{A},\mathbf{b},\mathbf{c}$ represent constraint matrix, constraint right term, and objective coefficient. For MILPs, an exact solution framework commonly used is B&B. This method first ignores the integer constraints to obtain and solve the relaxed problem at the root node. Subsequently, it iteratively searches for the global optimal solution through branching, bounding, and pruning. Branching involves selecting a variable with a fractional solution $x_j = b_j$ at the current node and adding the constraints $x_j \leq [b_j]$ and $x_j \geq [b_j] + 1$ to form two child nodes. During bounding, the global upper and lower bounds (also known as the primal and dual bounds) are determined based on all existing nodes. The pruning process eliminates obviously infeasible nodes according to these bounds. This procedure repeats until the upper and lower bounds converge, yielding the global optimal solution.

**Branching Rules.** Here, the branching variable selection during B&B significantly impacts solving efficiency by influencing tree size. In [2], several heuristic branching rules are introduced. Among them, strong branching evaluates candidate variables by creating child nodes for each candidate and selecting the one maximizing dual bound improvement. While highly effective, this approach incurs significant computational overhead that counteracts its benefits for solution speed. Pscost branching guides current branching by leveraging historical branching records, avoiding extra computation but performing poorly early in the search tree due to insufficient branching records. Hybrid approaches combine both methods' advantages by using strong branching initially to establish reliable branching patterns, then transitioning to pscost branching once sufficient historical data is accumulated. The state-of-the-art (SOTA) relpscost branching [1], SCIP's default rule, implements this strategy—a variable's pscost is only considered trustworthy after undergoing sufficient strong branching steps.

### 3.2 Parameterized B&B Tree

To parameterize the B&B tree and obtain instance-independent features for heterogeneous MILP problems, Zarpellon et al. [48] design a state representation $s_t = (C_t, Tree_t)$. Here, $C_t \in \mathbb{R}^{|\mathcal{C}_t|\times 25}$ denotes candidate variable features, and $Tree_t \in \mathbb{R}^{61}$ represents tree features, where $\mathcal{C}_t$ denotes candidate variable set and $|\mathcal{C}_t|$ represents candidate variable number. Since all features reflect the dynamic process of B&B trees, all MILP instances can be processed uniformly in the same feature space by a neural network named TreeGate, which jointly processes candidate and tree features through two components: a candidate network and a tree network. The candidate network first embeds each variable's 25-dimensional features into an $h$-dimensional space, then progressively reduces the dimensionality from $h$ to $d$ through multiple layers that halve the dimension at each step. Meanwhile, the tree network projects the tree features $Tree_t$ into an $H$-dimensional space (where $H = h + h/2 + \ldots + d$) using a sigmoid activation to produce a gating vector $g \in [0,1]^H$. This gating vector modulates the candidate network's layer outputs through element-wise multiplication. The final output $e_t \in \mathbb{R}^{|\mathcal{C}_t|\times d}$ undergoes average pooling across the $d$-dimensional features, then being processed by a softmax layer to generate the candidate variable selection probabilities.

## 4 Methodology

In this section, we formally introduce Mamba-Branching, a neural branching policy specifically designed to capture the sequential structure of B&B trees. We begin by discussing the contrastive learning approach utilized for the embedding layer and the detailed design of the sequence inputs, followed by the detailed implementation of imitation learning. The overall framework of Mamba-Branching is illustrated in Figure 1.

### 4.1 Contrastive Learning for Embedding Layer

The embedding layer serves as a critical interface between raw state representations and downstream sequence models. In natural language processing (NLP), the success of embedding techniques has been well-established. These methods leverage the inherent distinguishability of discrete word tokens, where each word's unique identity naturally translates to separable embedding vectors through

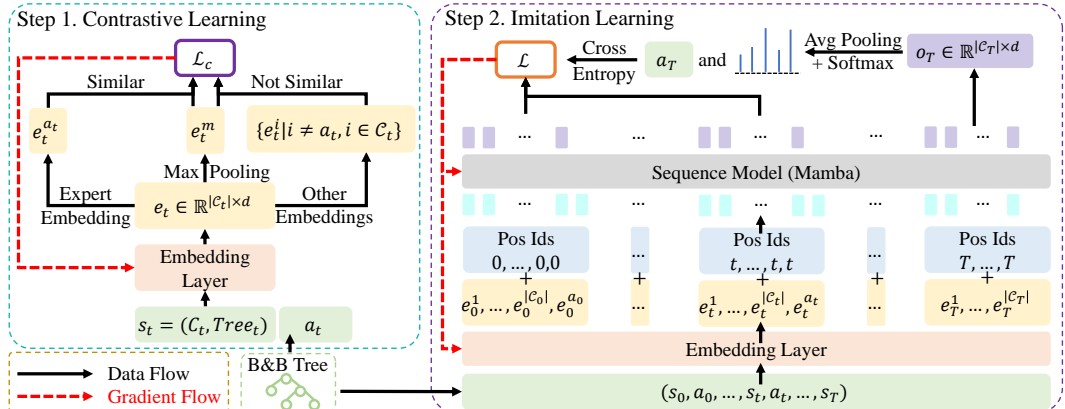

Figure 1: Overall framework of Mamba-Branching. The training process involves two stages: contrastive learning and autoregressive imitation learning. During the contrastive learning process, the state $s_t$ and expert decision $a_t$ at each branching step $t$ are used to train the embedding layer via the designed contrastive loss function $\mathcal{L}_c$. During imitation learning, the branching trajectory $(s_0, a_0, ..., s_T)$ is mapped to embeddings. At step $t$, expanded variable embeddings $(e_t^1, ..., e_t^{|\mathcal{C}_t|})$ and expert embedding $e_t^{a_t}$ form a group with shared positional encoding. These groups create a "branching path" input to the sequence model, where only outputs $o_t$ corresponding to the variable embeddings are selected, with $a_t$ serving as the label for imitation learning.

standard training paradigms. [4, 35] However, the branching variable selection problem in B&B presents a fundamentally different challenge. The state representation at each branching step $t$, denoted as $s_t = (C_t, Tree_t)$ (see subsection 3.2), contains a set of candidate variables $\mathcal{C}_t$ that frequently exhibit remarkably similar feature characteristics. This high degree of intra-step similarity arises from the shared constraints and problem structure inherent in combinatorial optimization problems. Unlike the clear distinctions between words in NLP tasks, the subtle but decision-critical differences between candidate variables in B&B require a more sophisticated approach to embedding learning.

To address this challenge, we develop a principled framework for learning discriminative embeddings in B&B decision making. The core of our approach lies in recognizing that effective branching decisions require the embedding space to maintain consistent separation between selected and non-selected variables. We formalize this requirement through Proposition 1. This condition specifies that the similarity between the selected variable's embedding and a reference vector must exceed all other candidate similarities by a positive margin $\delta$.

**Proposition 1.** *For effective branching decisions, the embedding space must satisfy:*

$$\forall t, \exists \delta > 0 \ s.t. \ sim(e_t^a, e_t^m) \geq \max_{i \neq a} sim(e_t^i, e_t^m) + \delta, \tag{1}$$

*where $sim(\cdot)$ is a similarity judgment function, $e_t$ denotes embeddings, $e_t^m$ is an anchor, $a$ denotes the selected variable index.*

To this end, before joint training, we first employ contrastive learning to train the embedding layer, enhancing its ability to differentiate between distinct candidate variable features. The loss function of contrastive learning is defined as $\mathcal{L}_c$, with the specific form as follows:

$$\mathcal{L}_c(\gamma) = \frac{1}{T} \sum_t \left( -\frac{e_t^m \cdot e_t^a}{\|e_t^m\|\|e_t^a\|} + \frac{1}{|\mathcal{C}_t| - 1} \sum_{i \neq a} \frac{e_t^m \cdot e_t^i}{\|e_t^m\|\|e_t^i\|} \right), \tag{2}$$

where $\gamma$ denotes the parameters of TreeGate, $T$ denotes the total number of branching steps, $e_t = \text{TreeGate}_\gamma(s_t)$, $e_t^m \in \mathbb{R}^d$ is the result of applying max-pooling to $e_t$ along the $|\mathcal{C}_t|$ dimension, $e_t^a \in \mathbb{R}^d$.

The intuition behind this loss function design is to make the selected branching variable the most prominent and distinctive among all candidate variables. The max-pooling operation extracts a salient global feature as an anchor. By increasing the cosine similarity between the anchor and the selected

branching variable while decreasing the cosine similarity between the anchor and other candidate variables, the loss amplifies their differences and drives the feature of the selected branching variable toward the globally most salient direction.

## 4.2 Sequential Modeling Design

In B&B tree, nodes are progressively expanded until the upper and lower bounds converge. This process can be viewed as navigating through a complex maze to find a "branching path" from the root node to the optimal solution node. Traditional neural approaches to branching decisions have predominantly relied on the immediate state of the tree, neglecting the historical sequence of visited nodes and prior branching choices. This myopic perspective is fundamentally limiting, as it fails to leverage the rich sequential information inherent in the branching process. Just as an effective maze-solving strategy requires reasoning about the entire traversed path to avoid dead ends and redundant exploration, optimal branching decisions demand a holistic understanding of the search trajectory. This underscores the imperative for a paradigm shift toward path-aware sequential modeling.

To effectively model branching decisions, the sequence model must capture not only the sequential progression of states but also the intricate interrelationships among candidate variables within each state. Therefore, we explicitly encode the features of each candidate variable at all branching steps. We formally define the branching path $\mathbf{S}$ as a structured sequence of embeddings, where each state at step $t$ is decomposed into its constituent candidate variables along with the selected branching decision. Specifically, $\mathbf{S}$ is represented as:

$$\mathbf{S} = [\underbrace{e_0^1, \ldots, e_0^{|\mathcal{C}_0|}, e_0^{a_0}}_{|\mathcal{C}_0|+1}, \ldots, \underbrace{e_t^1, \ldots, e_t^{|\mathcal{C}_t|}, e_t^{a_t}}_{|\mathcal{C}_t|+1}, \ldots, \underbrace{e_T^1, \ldots, e_T^{|\mathcal{C}_T|}}_{|\mathcal{C}_T|}], \tag{3}$$

where $|\mathcal{C}_t|$ denotes the number of candidate variables at branching step $t$, $a_t$ represents the index of the selected branching variable, $e_t$ denotes the embedding feature, and $T$ is the maximum number of branching steps in the branching path. This formulation ensures that both the sequential dynamics and the variable-level interactions are preserved, enabling the model to leverage granular features for improved decision-making.

To ensure temporal coherence across branching steps, we employ positional encodings that assign identical positional indices to embeddings within the same step. The complete input representation $\mathbf{S}'$ is constructed by combining the branching path $\mathbf{S}$ with a learnable positional encoding matrix $\mathbf{E}_{pos}$:

$$pos = [\underbrace{0, \ldots, 0, 0}_{|\mathcal{C}_0|+1}, \ldots, \underbrace{T, \ldots, T}_{|\mathcal{C}_T|}],$$
$$\mathbf{S}' = \mathbf{E}_{pos} \oplus \mathbf{S}, \tag{4}$$

where $\oplus$ denotes element-wise addition, $\mathbf{E}_{pos} \in \mathbb{R}^{(\sum_{t=0}^T |\mathcal{C}_t|+T) \times d}$ represents the learnable positional encoding matrix obtained by mapping $pos$.

Subsequently, $\mathbf{S}'$ is fed into Mamba to obtain the output $\mathbf{O}_t$:

$$\mathbf{O}_t = \text{Mamba}(\mathbf{S}')$$
$$= [o_0^1, \ldots, o_0^{|\mathcal{C}_0|}, o_0^{a_0}, \cdots, o_t^1, \ldots, o_t^{|\mathcal{C}_t|}, o_t^{a_t}, \cdots, o_T^1, \ldots, o_T^{|\mathcal{C}_T|}], \tag{5}$$

within each group, only the outputs corresponding to $|\mathcal{C}_t|$ variable positions are extracted, denoted as $o_t = (o_t^1, \ldots, o_t^{|\mathcal{C}_t|})$, which are then processed through average pooling and softmax to obtain the variable probability distribution.

It can be observed that for the branching path $\mathbf{S}$, the sequence model actually needs to process an input length of $\sum_{t=0}^T |\mathcal{C}_t| + T$. When either $T$ or $|\mathcal{C}_t|$ becomes large, the length of $\mathbf{S}$ increases substantially, presenting significant challenges to the sequence model's ability to handle long sequences. Therefore, in addition to employing the most commonly-used Transformer Decoder as our sequence model, we also utilize Mamba [14] (see Appendix A for architectural details). In contrast to the Transformer's quadratic complexity, Mamba achieves linear complexity relative to sequence length, making it unequivocally better suited for such long-sequence application scenarios. This is particularly critical in our application, where computational speed is paramount. If the neural branching policy's inference complexity becomes excessively high and computationally prohibitive, it would fundamentally undermine our original objective of acceleration.

### 4.3 Imitation Learning under Autoregressive Paradigm

Following prior works [48, 31], we employ relpscost branching as the expert to collect demonstration datasets for imitation learning. In contrast to the commonly employed strong branching expert [12, 15], which are rarely applied in practical scenarios, relpscost provides a more realistic expert representation. For dataset collection, each instance is solved using SCIP. We sequentially record every state in the instance's tree along with the corresponding relpscost-selected branching decisions, resulting in a complete trajectory denoted as $(s_0, a_0, s_1, a_1, \ldots)$. In dataset $\mathcal{D}$, each instance's trajectory is partitioned into fixed-length sub-trajectories for storage.

In Mamba-Branching, the branching policy is defined as $\pi_\theta$, which operates in an autoregressive paradigm. The joint loss function $\mathcal{L}(\theta, \gamma)$ of embedding layer and sequence model is as follows:

$$\mathcal{L}(\theta, \gamma) = -\frac{1}{|\mathcal{D}|} \sum_{\tau \in \mathcal{D}} \sum_{(s_t, a_t) \in \tau} \log \pi_\theta(a_t | \tau_{0:t}), \tag{6}$$

where $\tau$ denotes a trajectory in $\mathcal{D}$, $\tau_{0:t} = (s_0, a_0, \ldots, a_{t-1}, s_t)$, and $|\mathcal{D}|$ represents the total number of trajectories in $\mathcal{D}$. During inference, the predictions from previous branching steps serve as input for the current step, yielding the probability distribution $\pi_\theta(\cdot | \hat{\tau}_{0:t})$ over candidate variables, where $\hat{\tau}_{0:t} = (s_0, \hat{a}_0, \ldots, \hat{a}_{t-1}, s_t)$.

## 5 Experiments

### 5.1 Setup

#### 5.1.1 Benchmarks

**MILP dataset.** Our method is designed to maintain generalization capability across heterogeneous MILPs. Therefore, the training and test instances are deliberately constructed to be distinct, with the strict requirement that the test set should not contain any instances present in the training set. Following the selection of instances from previous works [48, 31], we construct two MILP datasets of different scales using instances from MIPLIB [13] and CORAL [29]: a smaller-scale dataset (MILP-S) and a larger-scale dataset (MILP-L). The MILP-S is entirely derived from [48], comprising 19 training instances and 8 test instances. MILP-L is constructed by expanding the dataset used in [31], containing 25 training instances and 73 test instances. For MILP-L's test instances, we employ SCIP as the reference solver, categorizing 57 instances with solution times under 20 minutes as "easy" and 16 instances exceeding 20 minutes as "difficult". The details of MILP-S and MILP-L are provided in Appendix B.

**Branching Dataset Collection.** During data collection, consistent with previous works [48, 31], we employ random branching for the first $r$ steps to enhance B&B exploration. After these $r$ random steps, we switch to relpscost branching and collect the corresponding data. For each training instance, we configure $r \in \{0, 1, 5, 10, 15\}$ and collect training set using solver seeds $\{0, 1, 2, 3\}$, while reserving seed 4 exclusively for validation set.

#### 5.1.2 Metrics

**Nodes and Fair Nodes.** The number of nodes in the B&B tree serves as a crucial metric for evaluating branching policies, as it directly impacts overall solving time. However, as noted in [10], this metric may be confounded by side effects of some sophisticated branching rules, such as strong branching. We therefore additionally employ the fair node number [10], which eliminates the confounding effects of these rules, thereby providing a more accurate reflection of the true capability of a branching policy. For branching policies that do not use strong branching, the number of nodes and fair nodes remains identical.

**Primal-Dual Integral.** For some challenging instances in MILP-L, obtaining optimal solutions may be computationally prohibitive, so a one-hour time limit is imposed. Under this constraint, node number becomes an inadequate metric for evaluating the performance of branching policies. In such cases, the primal-dual integral (PD integral) serves as a more appropriate evaluation criterion [11]. With a time limit $T_l$, the PD integral is expressed as $\int_{t=0}^{T_l} \mathbf{c}^\top \mathbf{x}_t^\star - \mathbf{y}_t^\star \, \mathrm{d}t$, where $\mathbf{y}_t^\star$ is the best dual bound at time $t$, $\mathbf{x}_t^\star$ is the best feasible solution at time $t$.

### 5.1.3 Baselines

We select two categories of branching policies as baselines: neural-based approaches and heuristic rules. The neural branching policies include: GCNN [12], TreeGate [48], T-BranT [31], and Transformer-Branching. GCNN is the most classical method and does not incorporate specific designs for heterogeneous MILPs. TreeGate and T-BranT are also based on instance-independent inputs, serving as the primary baselines for comparison with Mamba-Branching. Transformer-Branching employs Transformer as the sequence model to highlight Mamba's advantages. The heuristic rules include random, pscost, and relpscost, where random and pscost represent the lower bounds of performance, relpscost serves as the expert and constitutes the upper bound of branching performance. However, neural branching policies may surpass relpscost in solving efficiency. More detailed reasons for the selection of baselines can be found in Appendix C.

### 5.1.4 Solver and Neural Policy Settings

**Solver Settings.** In our evaluation, we replace SCIP solver's (v8.0.4) branching policy with our neural branching policy. To isolate the study of branching policies and eliminate interference from other solver components, we disable all primal heuristics and provide each test instance with a known optimal solution value as a cutoff. However, during branching data collection, we intentionally omit the cutoff to obtain longer branching sequences.

**Neural Policy Settings.** During Mamba-Branching training, the maximum branching step is $T = 99$, but as shown in subsection 4.2, its corresponding actual input length is considerably long. When using the Transformer as the sequence model, this length causes excessive GPU memory consumption that exceeds hardware limitations, thus $T = 9$ is adopted during Transformer-Branching training. For evaluation consistency, we employ autoregressive generation with $T = 24$ across all models. Additional implementation details and hyperparameters can be found in Appendix D.

Table 1: The experimental results on MILP-S. For the 8 test instances in MILP-S, each instance is evaluated with five random seeds $\{0,1,2,3,4\}$ under a 1-hour time limit, and the results are presented as geometric means. Among them, blue background indicates the best results, **bold** font indicates the best results in neural policies, and $\star$ denotes reaching the time limit.

| Method | Mamba-Branching | TreeGate | Transformer-Branching | T-BranT | GCNN | random | pscost | relpscost |
|---|---|---|---|---|---|---|---|---|
| Nodes | **2054.99** | 2171.31 | 3078.56 | 2668.62 | 33713.63$^\star$ | 61828.29$^\star$ | 4674.34 | 730.21 |
| Fair Nodes | **2077.55** | 2205.06 | 3120.04 | 2715.16 | 33713.63$^\star$ | 61828.29$^\star$ | 4674.34 | 1227.25 |

## 5.2 Branching Performance

### 5.2.1 MILP-S

The experimental results in MILP-S can be found in Table 1, with the fair node results of all neural branching policies per instance shown in Figure 2. The single-step inference time comparison between Transformer and Mamba is shown in Figure 3. Notably, T-BranT necessitates at least one set of historical data, prompting the use of relpscost at the root node. This precise branching decision at the root significantly influences overall performance. For the sake of consistency, Mamba-Branching, TreeGate, and Transformer-Branching also employ relpscost at the root node, with Mamba-Branching and Transformer-Branching further leveraging it to initialize their input sequences. To evaluate the performance of pure neural branching, we additionally test variants that do not utilize relpscost initialization: TreeGate-p, Mamba-Branching-p, and Transformer-Branching-p, as shown in Table 3.

It can be observed that Mamba-Branching is the best branching policy besides relpscost. First, Mamba-Branching significantly outperforms the three lower-bound references: GCNN, random, and pscost. Compared with several neural branching policy baselines, whether initialized with relpscost or purely neural-based, Mamba-Branching surpasses T-BranT, TreeGate, and Transformer-Branching, achieving a new SOTA for neural branching policies. Additionally, in terms of single-step inference time, Mamba significantly outperforms Transformer, highlighting its advantage as a sequence model.

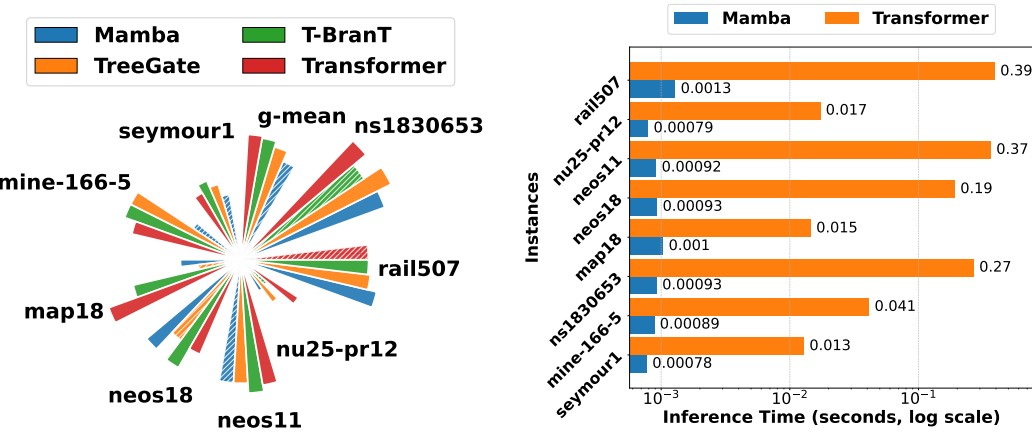

Figure 2: The fair node results of all neural branching policies in MILP-S.

Figure 3: The inference time comparison between Mamba and Transformer in MILP-S.

### 5.2.2 MILP-L

In MILP-L, we further evaluate several methods that demonstrated strong performance in MILP-S, including TreeGate, T-BranT, Mamba-Branching, and relpscost. For the 57 easy instances, performance is assessed using both nodes and fair nodes. In contrast, for the 16 difficult instances, we use the PD integral with a 1-hour time limit as the evaluation metric. The results are presented in Table 2.

The results demonstrate that for easy MILPs, despite the expanded test instances compared to MILP-S, Mamba-Branching remains the best neural branching policy,

Table 2: The experimental results on MILP-L. Consistent with the experiments on MILP-S, all instances are evaluated with 5 random seeds under a 1-hour time limit, with results reported as geometric means. Blue background indicates the best results and **bold** font indicates the best results in neural policies.

| Method | Easy | | Difficult |
|---|---|---|---|
| | Nodes | Fair Nodes | PD Integral |
| Mamba-Branching | **1819.32** | **2053.91** | **12319.55** |
| TreeGate | 2218.29 | 2534.09 | 14625.68 |
| T-BranT | 2009.77 | 2298.76 | 13538.47 |
| relpscost | 667.93 | 1455.49 | 12741.36 |

but still inferior to relpscost. For difficult instances, Mamba-Branching achieves the best PD integral performance among all branching policies, even surpassing relpscost. This indicates that within the same time limit, Mamba-Branching enables the fastest convergence of primal and dual bounds.

### 5.2.3 Discussion

**Advantage of Sequential Nature.** First, Mamba-Branching consistently outperforms TreeGate and T-BranT across all scenarios due to its consideration of the sequential nature of B&B trees. Neither TreeGate (which completely ignores historical data) nor T-BranT (which utilizes historical data non-sequentially) achieves the effectiveness of sequential historical data utilization. This aligns with our maze analogy in subsection 4.2: sequentially recalling paths facilitates better current decision-making.

**Limitation of Transformer.** Transformer-Branching also leverages sequential nature but performs poorly, with Transformer-Branching-p even underperforming the lower-bound pscost. The suboptimal performance stems from its 10-step branching history limit during training (due to hardware constraint), while Mamba-Branching accommodates 100 steps. Furthermore, the inference time comparison in Figure 3 demonstrates that in our time-sensitive scenario aimed at reducing solving time, Transformer is entirely unsuitable as a branching policy. The underlying reason here is that Transformer's complexity is quadratic with respect to sequence length, while Mamba's is linear. Although Transformer is theoretically suitable as a sequence model, employing Mamba offers greater practicality and feasibility. A more detailed comparison can be found in Appendix E.

**Factors Outperforming Relpscost.** As for relpscost, Mamba-Branching does not outperform in easy instances but surpasses it in difficult ones. The reason can be summarized as follows: (1) Relpscost is a hybrid method combining strong and pscost branching, incorporating a reliability criterion: a

Table 3: On MILP-S, the results of pure neural branching policies TreeGate-p, Transformer-Branching-p, and Mamba-Branching-p, as well as Mamba-Branching-p without contrastive learning (w/o cl). The experimental setup remains consistent with the aforementioned configuration on MILP-S.

| Method | Mamba-Branching-p | Mamba-Branching-p (w/o cl) | Transformer-Branching-p | TreeGate-p |
|---|---|---|---|---|
| Nodes | **2272.43** | 3000.92 | 5138.15 | 3179.55 |
| Fair Nodes | **2272.43** | 3000.92 | 5138.15 | 3179.55 |

variable can only switch to pscost after being selected by strong branching a certain number of times. Therefore, for difficult instances with more variables, the initialization process is time-consuming, leading to potential inefficiency. In contrast, neural policies benefit from fast inference and exhibit advantages on difficult instances. (2) In relpscost, the use of pscost for leveraging historical data does not account for the sequential nature, whereas Mamba-Branching explicitly incorporates this consideration. (3) As mentioned in [48], the relpscost in SCIP has been fine-tuned for a large number of instances, resulting in excellent performance on easy instances. However, for more complex and challenging instances, such parameter tuning may not provide adequate coverage.

### 5.3 Ablation Study

In this section, an ablation experiment is conducted to verify the role of contrastive learning. First, the most straightforward comparison is to evaluate the branching performance difference when contrastive learning is applied or not to the embedding layer. Under pure neural branching, the results on MILP-S without and with contrastive learning are denoted as Mamba-Branching-p (w/o cl) and Mamba-Branching-p, respectively, as shown in Table 3. Meanwhile, to demonstrate that contrastive learning indeed achieves its intended effect, that is, making the feature of expert-selected variable more distinguishable compared to other candidates, we also visualize the t-SNE-reduced [34] embeddings, as shown in Figure 4.

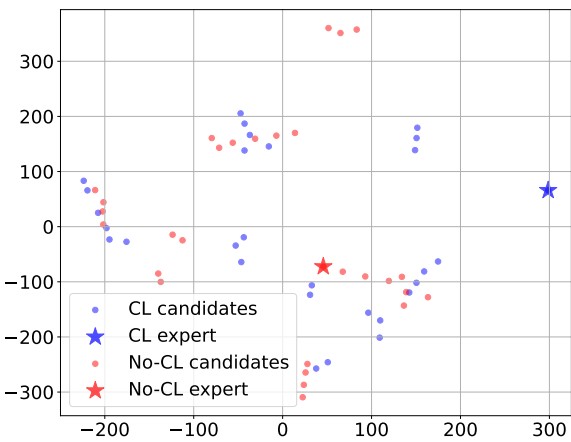

Figure 4: At a random given state, the embeddings with and without contrastive learning.

First, in terms of branching performance, Mamba-Branching-p demonstrates superior results compared to its counterpart without contrastive learning, Mamba-Branching-p (w/o cl). The visualization then reveals that with contrastive learning, the expert-selected variable exhibits greater outlier characteristics and enhanced discriminability relative to other candidate variables. In contrast, without contrastive learning, the expert-selected variable becomes less distinguishable and tends to cluster near candidate variables.

## 6 Conclusion and Future Work

In this paper, we propose Mamba-Branching, the first approach to consider the sequential nature in B&B trees. To address the challenges of long sequences and embedding distinctiveness posed by sequential nature, we employ Mamba as the sequence model and design a contrastive learning method to train the embedding layer, enabling the sequence model to distinguish between different candidate variables. In experiments, Mamba-Branching outperforms all neural branching policies and achieves superior solving efficiency compared to relpscost on challenging instances. One limitation of our approach is the reliance on imitation learning, which requires a time-consuming collection of expert demonstrations. In future work, we will focus on investigating the potential of sequential nature in reinforcement learning-based branching policies, thereby eliminating the dependency on expert data.

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

## A  Mamba Architecture

Mamba is a novel network architecture based on State Space Model (SSM) that may potentially replace self-attention-based Transformer models [14, 8, 44]. SSM is a concept originating from control theory, with its earliest roots traceable to the classical Kalman filter [23]. The continuous-time formulation of SSM can be represented as follows:

$$
\begin{aligned}
\dot{\mathbf{z}}(t) &= \mathbf{M}(t)\mathbf{z}(t) + \mathbf{N}(t)\mathbf{u}(t) \\
\mathbf{y}(t) &= \mathbf{P}(t)\mathbf{z}(t),
\end{aligned}
\tag{7}
$$

where $\mathbf{z}(t) \in \mathbb{R}^z$, $\mathbf{y}(t) \in \mathbb{R}^q$, $\mathbf{u}(t) \in \mathbb{R}^u$. After zero-order hold discretization, the discrete-time formulation is obtained as follows:

$$
\begin{aligned}
\mathbf{z}_t &= \overline{\mathbf{M}}\mathbf{z}_{t-1} + \overline{\mathbf{N}}\mathbf{u}_t \\
\mathbf{y}_t &= \mathbf{P}\mathbf{z}_t,
\end{aligned}
\tag{8}
$$

where $\overline{\mathbf{M}} = \exp(\Delta\mathbf{M})$, $\overline{\mathbf{N}} = (\Delta\mathbf{M})^{-1}(\exp(\Delta\mathbf{M}) - \mathbf{I}) \cdot \Delta\mathbf{N}$, $\Delta$ denotes the step size.

To meet the parallelization requirements of the training process, the SSM can alternatively be represented as follows:

$$
\begin{aligned}
\overline{\mathbf{K}} &= (\mathbf{P}\overline{\mathbf{N}}, \mathbf{P}\overline{\mathbf{M}}\overline{\mathbf{N}}, \dots, \mathbf{P}\overline{\mathbf{M}}^k\overline{\mathbf{N}}) \\
\mathbf{y} &= \overline{\mathbf{K}} * \mathbf{u}.
\end{aligned}
\tag{9}
$$

Mamba builds upon SSM by introducing Selective SSM, which essentially treats $\mathbf{N}$, $\mathbf{P}$ and $\Delta$ as functions of the input while keeping M unchanged. From a control theory perspective, this transforms the system from time-invariant to time-varying. Furthermore, Mamba incorporates hardware-aware algorithm design that enables efficient storage of intermediate results through parallel scanning, kernel fusion, and recalculation.

## B  Benchmark Details

All training and test instances in MILP-S are listed in Table 6. The training instances in MILP-L are presented in Table 7. All easy test instances in MILP-L are shown in Table 8, while all difficult test instances are presented in Table 9. Here, an instance is considered easy if SCIP's solving time is less than 20 minutes; otherwise, it is classified as difficult. These instances are sourced from MIPLIB [13] and CORAL [29], all collected from real-world application scenarios, with specific instance selections referenced to [48] and [31]. Serving as benchmarks, these instances effectively reflect the practical significance of neural branching policies in real-world applications.

## C  Detailed reasons for Baseline Selection

The neural branching policies and their selection rationale are detailed below: (1) GCNN [12]: This method is not designed for heterogeneous MILPs and performs poorly on unseen MILP instances outside the training distribution. The experimental results of GCNN highlight the advantage of instance-independent feature design in terms of generalization. (2) TreeGate [48]: The TreeGate network incorporates instance-independent inputs by design, making it suitable for heterogeneous MILPs. Additionally, since Mamba-Branching's embedding layer adopts the TreeGate architecture, TreeGate serves as a critical control group for our method. (3) T-BranT [31]: Building upon Tree-Gate's feature design, T-BranT employs attention to capture mutual connections among candidates. Meanwhile, T-BranT processes historical data from an unordered graph perspective. The comparison with T-BranT serves to evaluate whether Mamba-Branching's sequential processing of historical data demonstrates superior performance over T-BranT's unordered graph approach. (4) Transformer-Branching: When selecting a sequence model for our approach, the Transformer would naturally be the most immediate consideration. Thus, we include Transformer-Branching as a comparative baseline against Mamba-Branching, specifically to highlight the advantages of employing Mamba as the sequence model.

The heuristic rules are selected with the following rationale: (1) Random: Serves as the performance lower bound, demonstrating the detrimental effects of completely omitting a deliberate branching

policy. (2) Pscost: A purely historical data-driven branching method that, like Random, also establishes a performance lower bound. (3) Relpscost: The expert policy in the imitation learning of Mamba-Branching. Simultaneously, it is also the SOTA heuristic rule and the default rule in SCIP. Relpscost serves as the upper bound of decision accuracy for neural branching policies. However, benefiting from the fast inference speed of neural networks, neural branching policies may surpass relpscost in terms of efficiency.

# D    Implementation Details

All experiments in this paper are run on NVIDIA A100-PCIE-40GB GPU and Intel(R) Xeon(R) Gold 5218 CPU. The hyperparameters in the training of Mamba-Branching are shown in Table 4.

During training, sequences consisting of every 100 branching steps are fed as input to Mamba. Although the number of candidate variables varies across batches, several hundred candidates represent a typical scenario. Consequently, the actual input sequence length may extend to tens of thousands of tokens – a scale that would easily trigger GPU memory overflow if using Transformer architectures. We therefore adopt Mamba as our sequence model, whose computational complexity scales linearly with sequence length, to effectively circumvent these hardware limitations.

During inference, the sequence fed into Mamba consists of: (1) states and predicted actions from the most recent 24 branching steps, and (2) the current state. Unlike the training phase, we deliberately reduce the number of branching steps to maintain sufficiently short inference latency, thereby reducing the total solving time. After systematic tuning, we ultimately select 25 branching steps (T_eva=24) to achieve an optimal balance between branching accuracy and inference speed.

Table 4: The hyperparameters of Mamba-Branching

| Name | Description | Value |
|---|---|---|
| d | Output dimension of the candidate net in TreeGate, which is equivalent to the embedding size of Mamba. | 8 |
| h | Hidden state dimension of the Candidate Net in TreeGate. | 64 |
| depth | Layer number of Tree Net in TreeGate. | 3 |
| batch_size | Batch size of Mamba Training | 32 |
| lr_cl | Learning rate of contrastive learning. | 0.0001 |
| optimizer_cl | Optimizer of contrastive learning. | Adam |
| lr | Learning rate of imitation learning. | 0.001 |
| optimizer | Optimizer of imitation learning. | AdamW |
| wd | Weight decay coefficient of imitation learning. | 0.01 |
| T_train | Maximum branching steps considered during training. | 99 |
| T_eva | Maximum branching steps considered during evaluating. | 24 |
| d_state | SSM state expansion factor in Mamba | 64 |
| d_conv | Local convolution width in Mamba | 4 |
| expand | Block expansion factor in Mamba | 2 |

# E    Computational Complexity Comparison between Transformer and Mamba

As is well-known, Mamba exhibits linear complexity with respect to sequence length, while Transformer demonstrates quadratic complexity. In this section, we present experimental results that provide a detailed comparison of the complexity between Mamba and Transformer when employed as branching policies. Our complexity analysis focuses on two key aspects: space complexity and time complexity. For space complexity, we compare the GPU memory consumption of Mamba and Transformer during both training and inference phases. Regarding time complexity, we primarily examine the inference latency of both models when functioning as branching policies. The experimental results are shown in Table 5.

As shown in the experiments, when processing 100 branching steps during training (even with a batch size of 1), Transformer-Branching fails to train altogether, while Mamba-Branching occupies minimal GPU memory. During inference, Mamba-Branching demonstrates significantly lower GPU

memory consumption and inference time. In contrast, Transformer-Branching not only requires substantially more GPU memory but, more critically, suffers from prohibitively long inference times. Since the fundamental purpose of adopting a neural branching policy is to accelerate MILP solving, such excessive inference time directly contradicts our original objective.

Table 5: A comparison of computational complexity between Mamba-Branching and Transformer-Branching. During training, we uniformly set T_train=99 with a batch size of 1. For inference, we consistently use T_eva=24. After collecting 25 Branching steps, we measure the network's inference time and GPU memory consumption, take the geometric mean across all test instances of MILP-S.

| Method | GPU memory of Train (GB) | GPU memory of Inference (GB) | Inference Time (s) |
|---|---|---|---|
| Mamba-Branching | 0.017 | 0.013 | 0.00093 |
| Transformer-Branching | out of memory | 1.051 | 0.075 |

Table 6: All instances in MILP-S.

| Instance | Variables | Constraints | Set |
|---|---|---|---|
| air04 | 8904 | 823 | train |
| air05 | 7195 | 426 | train |
| dcmulti | 548 | 473 | train |
| eil33-2 | 4516 | 32 | train |
| istanbul-no-cutoff | 5282 | 20346 | train |
| l152lav | 1989 | 97 | train |
| lseu | 89 | 28 | train |
| misc03 | 160 | 96 | train |
| neos20 | 1165 | 2446 | train |
| neos21 | 614 | 1085 | train |
| neos-476283 | 11915 | 10015 | train |
| neos648910 | 814 | 1491 | train |
| pp08aCUTS | 240 | 246 | train |
| rmatr100-p10 | 7359 | 7260 | train |
| rmatr100-p5 | 8784 | 8685 | train |
| sp150x300d | 600 | 450 | train |
| stein27 | 27 | 118 | train |
| swath1 | 6805 | 884 | train |
| vpm2 | 378 | 234 | train |
| map18 | 164547 | 328818 | test |
| mine-166-5 | 830 | 8429 | test |
| neos11 | 1220 | 2706 | test |
| neos18 | 3312 | 11402 | test |
| ns1830653 | 1629 | 2932 | test |
| nu25-pr12 | 5868 | 2313 | test |
| rail507 | 63019 | 509 | test |
| seymour1 | 1372 | 4944 | test |

Table 7: Training instances in MILP-L

| Instance | Variables | Constraints |
|---|---|---|
| 30n20b8 | 18380 | 576 |
| air04 | 8904 | 823 |
| air05 | 7195 | 426 |
| cod105 | 1024 | 1024 |
| comp21-2idx | 10863 | 14038 |
| demulti | 548 | 290 |
| eil33–2 | 4516 | 32 |
| istanbul-no-cutoff | 5282 | 20346 |
| l152lav | 1989 | 97 |
| lseu | 89 | 28 |
| misc03 | 160 | 96 |
| neoS20 | 1165 | 2446 |
| neoS21 | 614 | 1085 |
| neos-476283 | 814 | 1491 |
| neos648910 | 11915 | 10015 |
| pp08aCUTS | 240 | 246 |
| rmatr100-p10 | 8784 | 8685 |
| rmatr100-p5 | 7359 | 7260 |
| rmatr200-p5 | 37816 | 37617 |
| roi5alpha10n8 | 106150 | 4665 |
| sp150 × 300d | 600 | 450 |
| stein27 | 27 | 118 |
| supportcase7 | 138844 | 6532 |
| swath1 | 6805 | 884 |
| vpm2 | 378 | 234 |

Table 8: Easy test instances in MILP-L , SCIP's solving time is less than 20 minutes.

| Instance | Variables | Constraints | SCIP Solving Time |
|---|---|---|---|
| aflow40b | 2728 | 1442 | 375.32 |
| app1-2 | 26871 | 53467 | 662.56 |
| bc1 | 1751 | 1913 | 237.68 |
| bell3a | 133 | 123 | 1.54 |
| bell5 | 104 | 91 | 0.63 |
| biella1 | 7328 | 1203 | 271.30 |
| binkar10_1 | 2298 | 1026 | 47.79 |
| blend2 | 353 | 274 | 0.37 |
| dano3_5 | 13873 | 3202 | 189.77 |
| fast0507 | 63009 | 507 | 150.11 |
| map10 | 164547 | 328818 | 515.00 |
| map18 | 164547 | 328818 | 250.49 |
| map20 | 164547 | 328818 | 218.78 |
| mik-250-20-75-4 | 270 | 195 | 55.67 |
| mine-166-5 | 830 | 8429 | 36.83 |
| misc07 | 260 | 212 | 28.94 |
| n2seq36q | 22480 | 2565 | 497.79 |
| neos11 | 1220 | 2706 | 171.35 |
| neos12 | 3983 | 8317 | 674.23 |
| neos-1200887 | 234 | 633 | 16.46 |
| neos-1215259 | 1601 | 1236 | 110.12 |
| neos13 | 1827 | 20852 | 96.20 |
| neos18 | 3312 | 11402 | 27.63 |
| neos-4722843-widden | 77723 | 113555 | 864.13 |
| neos-4738912-atrato | 6216 | 1947 | 304.32 |
| neos-480878 | 534 | 1321 | 52.92 |
| neos-504674 | 844 | 1344 | 114.11 |
| neos-504815 | 674 | 1067 | 34.61 |
| neos-512201 | 838 | 1337 | 42.39 |
| neos-584851 | 445 | 661 | 7.11 |
| neos-603073 | 1696 | 992 | 269.58 |
| neos-612125 | 9554 | 1795 | 43.31 |
| neos-612162 | 9893 | 1859 | 40.85 |
| neos-662469 | 18235 | 1085 | 566.6 |
| neos-686190 | 3660 | 3664 | 61.03 |
| neos-801834 | 3220 | 3300 | 51.71 |
| neos-803219 | 640 | 901 | 32.99 |
| neos-807639 | 1030 | 1541 | 20.52 |
| neos-820879 | 9522 | 361 | 56.56 |
| neos-829552 | 40971 | 5153 | 353.99 |
| neos-839859 | 1975 | 3251 | 64.10 |
| neos-892255 | 1800 | 2137 | 54.07 |
| neos-950242 | 5760 | 34224 | 149.60 |
| ns1208400 | 2883 | 4289 | 111.91 |
| ns1830653 | 1629 | 2932 | 148.27 |
| nu25-pr12 | 5868 | 2313 | 22.01 |
| nw04 | 87482 | 36 | 42.85 |
| p0201 | 201 | 133 | 0.81 |
| pg | 2700 | 125 | 45.39 |
| pp08a | 240 | 136 | 1.44 |
| rai507 | 63019 | 509 | 160.55 |
| roll3000 | 1166 | 2295 | 44.27 |
| rout | 556 | 291 | 39.51 |
| satellites1-25 | 9013 | 5996 | 952.04 |
| seymour1 | 1372 | 4944 | 61.91 |
| sp98ir | 1680 | 1531 | 92.98 |
| unitcal_7 | 25755 | 48939 | 889.85 |

Table 9: Difficult test instances in MILP-L , SCIP's solving time is more than 20 minutes.

| Instance | Variables | Constraints | SCIP Solving Time |
|---|---|---|---|
| atlanta-ip | 48738 | 21732 | 3600.00 |
| bab5 | 21600 | 4964 | 2665.41 |
| harp2 | 2993 | 112 | 1642.38 |
| map16715-04 | 164547 | 328818 | 2423.74 |
| msc98-ip | 21143 | 15850 | 3600.00 |
| mspp16 | 29280 | 561657 | 2722.23 |
| n3seq24 | 119856 | 6044 | 3600.00 |
| pigeon-10 | 490 | 931 | 3600.01 |
| bab2 | 147912 | 17245 | 3600.02 |
| bab6 | 114240 | 29904 | 3600.01 |
| neos-4338804-snowy | 1344 | 1701 | 3600.08 |
| neos-4387871-tavua | 4004 | 4554 | 3600.00 |
| neos-4647030-tutaki | 12600 | 8382 | 3600.09 |
| nursesched-medium-hint03 | 34248 | 14062 | 3600.00 |
| opm2-z10-s4 | 6250 | 160633 | 3600.01 |
| radiationm40-10-02 | 172013 | 173603 | 3600.00 |

