# OpenReview forum: "Towards Better Branching Policies: Leveraging the Sequential Nature of Branch-and-Bound Tree"
_NeurIPS.cc/2025/Conference — Submitted to NeurIPS 2025_

### Official Review · Reviewer_TcMy · 2025-06-18

**Clarity:** 3
**Significance:** 2
**Originality:** 2
**Rating:** 4
**Confidence:** 4

**Summary:**

The paper presents Mamba-Branching, a deep-learning-based branching strategy designed to enhance the Branch-and-Bound (B&B) algorithm for solving Mixed-Integer Linear Programming (MILP) problems. The key innovations include treating the branching path as a sequence to capture the influence of past decisions, using the Mamba architecture for efficient long-sequence processing, and applying contrastive learning to pre-train variable embeddings. The empirical results demonstrate significant improvements over existing neural branching strategies and even outperform SCIP’s default heuristic on challenging instances.

**Questions:**

1. **Notation Clarification (Lines 197–198):** Could the authors confirm whether \(o_t\) should indeed be \(o_T\) in the definition provided in Lines 197–198?
2. **Proof of Proposition:** Could the authors explain how to derive Proposition 1?
2. **Inconsistent Baseline Performance (Table 1):** T‑BranT’s results are consistently worse than TreeGate’s, which seems to contradict the numbers reported in Tables 6–7 of "Learning to Branch with Tree‑aware Branching Transformers”?
3. **Missing Ablation Data (Table 3):** Why is T-BranT’s performance omitted from the ablation table, and could the authors provide this data to facilitate a more comprehensive comparison?
4. **Incomplete Ablations.**
    Only the presence or absence of contrastive learning is tested. It would be informative to also evaluate:
   - Contrastive learning with standard (non‑Mamba) sequence modeling, and
   - The specific effect of positional encodings within the sequence‑modeling module.

**Ethical Concerns:**

["NO or VERY MINOR ethics concerns only"]

**Final Justification:**

I have carefully reviewed the author’s rebuttal and taken into account all the points raised. I am inclined to increase the score by 1 point.

**Limitations:**

- The paper’s claims are primarily validated on two specific MILP test sets (MILP-S and MILP-L), which may limit the generalizability of the results to other types of MILP instances.
- The paper does not explore the potential of reinforcement learning-based approaches, which could offer a different set of advantages and challenges in the context of branching policies.

**Quality:**

3

**Strengths And Weaknesses:**

### Strengths
- The paper systematically incorporates the sequential nature of B&B branching paths into a deep model, supported by clear theoretical motivation.
- The introduction of the linear-complexity Mamba architecture accelerates the branching process.
- Mamba-Branching shows consistent improvements across multiple metrics on the MILP-S and MILP-L test sets, even surpassing SCIP’s default relpscost heuristic on hard instances.
- The paper provides detailed descriptions of the contrastive learning and sequence-modeling modules, each validated via ablation studies.

### Weaknesses
- The ablation table (Table 3) omits T-BranT’s performance, making direct comparison difficult.
- The ablation studies are limited in scope and could be expanded to include evaluations of contrastive learning with standard sequence modeling and the effect of positional encodings within the sequence-modeling module.
- The paper does not compare Mamba-Branching against other recent sequence-aware neural architectures, such as those introduced in “Exact Combinatorial Optimization with Temporo-Attentional Graph Neural Networks,” leaving a gap in the evaluation of its relative merits.
- The paper acknowledges that Mamba-Branching does not universally outperform SCIP’s default relpscost heuristic, particularly on simpler instances, indicating that the approach may be more suitable for certain types of MILP problems.

---

> ### Author Rebuttal · Authors · 2025-07-29
>
> First, we appreciate you for highlighting the strengths of our paper: **incorporating the sequential nature of B&B, the introduction of the linear-complexity Mamba architecture, the detailed description of sequence modeling and contrastive learning along with ablation studies**, as well as your recognition of **the superior performance of our method on different datasets**.
> Next, please allow us to address the weaknesses and questions you raised one by one.
>
> > ### Response to Weaknesses 1 and Question 4: Missing Ablation of T-BranT-p
>
> In the original paper (line 284), it is explained that T-BranT must use relpscost at the root node once to obtain historical information. Therefore, there is no purely neural network-based T-BranT-p **unless their algorithm design is modified** (i.e., their source code needs to be changed). However, we have still attempted this approach, and the results are as follows.
>
> |            | Mamba-Branching-p | T-BranT-p |
> |------------|-------------------|-----------|
> | Nodes      | 2272.43           | 3922.63   |
> | Fair Nodes | 2272.43           | 3922.63   |
>
> Table 1: Comparison between Mamba-Branching-p and T-BranT-p on MILP-S. This table serves as a supplement to the ablation experiments. Here, "-p" denotes the pure neural network method without relpscost initialization.
>
> As can be observed, **T-BranT-p performs worse than Mamba-Branching-p**.
>
> > ### Response to Weakness 2 and Question 5: Incomplete Ablations -- Standard Sequence Modeling and Effect of Positional Encodings
>
> Thank you for your suggestion! We have already included other sequence modeling approach -- **Transformer-Branching-p** -- in our ablation study. Additionally, since positional encoding is a natural component in sequence models, it was initially overlooked in our ablation experiments. However, we are happy to provide further ablation results, including:
>
> 1. Testing GRU as an alternative sequence model, named **GRU-Branching-p** (*-p* denotes pure neural network), in addition to Transformer.
> 2. Results from Mamba-Branching-p with positional encoding removed, named **Mamba-Branching-p (w/o pe)**.
>
> The corresponding results are presented in the table below.
>
> |            | Mamba-Branching-p | Transformer-Branching-p | GRU-Branching-p | Mamba-Branching-p (w/o pe) |
> |------------|-------------------|-------------------------|-----------------|----------------------------|
> | Nodes      | 2272.43           | 5138.15                 | 5683.86         | 3481.19                    |
> | Fair Nodes | 2272.43           | 5138.15                 | 5683.86         | 3481.19                    |
>
> Table 2: Supplementary ablation experiments on MILP-S. All methods adopt pure neural networks without relpscost initialization. Among them, the method using Transformer as the sequence model has been provided in the original paper, and we further supplement the method using **GRU** as the sequence model. Meanwhile, the results **without positional encoding (w/o pe)** in Mamba-Branching-p is also presented.
>
> From Table 2, it can be observed that **Mamba demonstrates** superior performance as a sequence model in the branching policy compared to both **Transformer and GRU**. Additionally, the performance of Mamba-Branching degrades when positional encoding is removed, which confirms the **importance of positional encoding**.
>
> > ### Response to weakness 3: No Comparison with “Exact Combinatorial Optimization with Temporo-Attentional Graph Neural Networks”
>
> Thank you very much for pointing out this paper! We acknowledge that there is some similarity between their work and ours. However, their approach primarily focuses on **graph structures and graph neural networks (GNNs).** Relying on graph structures i**nherently limits generalization capability**, as it is only applicable to scenarios **involving repeatedly solving similar problems**. In contrast, **Mamba-Branching** is designed to handle **arbitrary heterogeneous problems**, which represents a key **distinction**.
>
> For further discussion, please refer to our responses to **Reviewer 3WGQ**, specifically the section titled **"Response to weakness 4: Why do we not use Ecole-Generated Instances nor compare with other baselines?"**:
>
> > ### Response to weakness 4: Suitable for Certain MILPs
>
> While our method does not universally outperform **relpscost**, it demonstrates superior performance on **real-world complex problems**. Compared to simpler cases, these complex problems better reflect **practical application** scenarios and offer higher **utility value**.
>
> > ### Response to Question 1: Notation Clarification
>
> The notation $o_t$ refers to the set of outputs at an arbitrary branching step t, while $o_T$ specifically denotes the set of outputs at the final branching step T. A more rigorous formulation would be: $\forall t \in\{0,1,...,T\}$.
>
> > ### Response to Question 2: Proof of Proposition
>
> Regarding Proposition 1, we will provide explanations from two perspectives: first, an intuitive explanation from the **conceptual standpoint**; second, an attempt to provide an explanation using **mathematical formulations**.
>
> First, we make the following two assumptions, which will serve as the foundation for our subsequent analysis:
>  1. The feature embeddings of the optimal branch decision variable ($e_t^a$) can be distinguished from those of other branch variables ($e_t^i, \forall i\in \mathcal{C}_t, i\neq a$) in the vector space.
> 2. The feature embedding $e_t$ is informative for branch decision-making.
>
> > #### Intuitive Explanation
>
> Based on assumption 1, we establish an anchor point $e_t^m$ and employ contrastive learning to train the model such that: (1) the optimal branch variable's feature $e_t^a$ is pulled closer to $e_t^m$, while (2) other variables' features are pushed away from $e_t^m$. This approach amplifies the separation between optimal and non-optimal features, thereby ensuring sufficient **distinguishability** when these features are subsequently processed by the Mamba architecture.
>
> > #### Mathematical Formulations
>
> For each branching variable $i$, we define $F(i)$ as the branching score function (in this paper, the **relpscost** score) for variable $i$, such that the optimal branching variable satisfies $$a = \arg\max_{i} F(i).$$
>
> Assumption 2 posits that the embeddings can encode branching quality information, which we simplify to a linear relationship $$e_t^i = \alpha F(i) + \epsilon_i,$$where $\alpha \in \mathbb{R}^d$ is a variable independent vector representing the quality-sensitive direction in the embedding space. $\epsilon_i \in \mathbb{R}^d$​ is a noise term satisfying $\mathbb{E}(\epsilon_i)=0$ with small magnitude $|\epsilon_i|$.
>
> We use max pooling over the embeddings as the anchor point, defined as:
> $$e_t^m = [...,\max_i e_t^i(j),...]
> =[...,\max_i (F(i)\alpha(j)+\epsilon_i(j)),...].$$
> For $e_t^a$, we have:
> $$e_t^a=[...,\max_i F(i)\alpha(j) + \epsilon_a(j),...].$$
> Indeed, if we assume that all $\epsilon_i$ terms are sufficiently similar in magnitude, we can derive the approximation $e_t^a \approx e_t^m$. Therefore, we can reasonably establish that
> $$\forall i \neq a,\text{sim}(e_t^a, e_t^m) > \text{sim}(e_t^i, e_t^m).$$
> This formulation is mathematically **equivalent** to the expression presented in the original paper.
> $$\forall t, \exists \delta > 0 \text{ s.t. } \text{sim}(e_t^a, e_t^m) \geq \max_{i \neq a} \text{sim}(e_t^i, e_t^m) + \delta$$
> Certainly, we acknowledge that the proof involves certain assumptions and approximations, which may lack full mathematical rigor. Additionally, during the neural network training process under contrastive loss, we cannot strictly guarantee that $\delta>0$ holds completely.
> However, we believe these results effectively **reflect the underlying intuition** behind our design: to make the optimal variable's features closer to the anchor point while pushing other features farther away, thereby making the optimal variable's features more **distinctive** in the vector space.
>
> > ### Response to Question 3: Inconsistent Baseline Performance
>
> In the original paper's Table 1, the results were obtained on the MILP-S dataset, which **differs from the dataset used in the original T-BranT paper (their dataset was more aligned with MILP-L)**. The MILP-S contains only **8** test cases and serves merely as a toy dataset for preliminary validation. On the **MILP-L** dataset, T-BranT outperforms TreeGate on both the **57** easy instances and **16** hard instances (see Table 2 at line 297 in our original paper). These results are **not contradictory** to those reported in the original T-BranT paper.
>
> > ### Response to limitation 1: Two Specific MILP Test Sets
>
> In fact, the number of test instances in our dataset **already exceeds** that of previous works (we have 57 + 16 = 73, while TreeGate [1] has 8 and T-BranT [2] has 66), which already demonstrates stronger generalizability.
>
> For more information, Please refer to the **Response to Weakness 1 to Reviewer KMMV: Insufficient experimental scale and lack of statistical significance analysis**.
>
> > ### Response to limitation 2: Potential of RL-based Methods
>
> Please refer to the following responses for **Reviewer 3WGQ, "Response to Weakness 2: Optimization Process vs. Supervised Imitation"**.
>
> To summarize, due to challenges such as sparse rewards and credit assignment, to the best of our knowledge, **RL paradigms have not yet surpassed imitation learning for branching policies**.
>
> > ### Reference
>
> [1] Zarpellon, Giulia, et al. "Parameterizing branch-and-bound search trees to learn branching policies." Proceedings of the aaai conference on artificial intelligence. Vol. 35. No. 5. 2021.
>
> [2] Lin, Jiacheng, et al. "Learning to branch with Tree-aware Branching Transformers." Knowledge-Based Systems 252 (2022): 109455.

---

> > ### Comment · Reviewer_TcMy · 2025-08-06
> >
> > Thank you for your thorough responses and the detailed experimental analyses. I've gained valuable insights from your rebuttal.
> >
> > I have an additional question regarding comparative analysis. In Appendix A.10 of the LIMIP paper [1], the authors report that LIMIP achieves approximately half the solve time and search tree size of TreeGate when evaluated on heterogeneous MILP instances. Have you conducted any direct comparisons between LIMIP and Mamba-Branching under similar heterogeneous problem settings?
> >
> > Specifically:
> > 1. Have you evaluated LIMIP against Mamba-Branching on comparable heterogeneous MILP benchmarks?
> > 2. If so, could you share the results regarding solve time and search tree size?
> > 3. If not, could you discuss the potential strengths and weaknesses of LIMIP compared to Mamba-Branching for heterogeneous MILP problems?
> >
> >
> > [1] LIMIP: Lifelong Learning to Solve Mixed Integer Programs, AAAI-2023. (https://arxiv.org/pdf/2208.12226)

---

> > > ### Author Response · Authors · 2025-08-06
> > > **Clarification Regarding Whether LIMIP Is Applicable to Heterogeneous Instances**
> > >
> > > Thank you for your response! We have read the paper you mentioned (**LIMIP** [1]), and their experimental setup for comparison against TreeGate is as follows, as stated **in the original text**:
> > >
> > > _"Comparison against TreeGate (Zarpellon et al. 2020): Comparison of Imitation learning of Strong Branching against TreeGate, which learns the weaker heuristic of reliability pseudo-cost branching. **Both models were trained and tested on SC0.05**."_
> > >
> > > We can confirm that they **"trained and tested on SC0.05,"** meaning their training and testing were conducted on **the same homogeneous instance** type -- SC0.05 -- which is fundamentally **different from the heterogeneous instances** used in our work. In fact, we trained on some instances and tested on entirely heterogeneous instances -- specifically, instances of entirely different types compared to the training set.
> > >
> > > We would like to reiterate the following **distinction**:
> > > Training and testing on the same problem type, with only parameter variations, is referred to as a **homogeneous** setting. In contrast, testing on instances with **completely different types and structures** from the training set constitutes a **heterogeneous** setting."
> > >
> > > As for why TreeGate underperforms LIMIP on homogeneous instances, we believe this is entirely expected: TreeGate disregards instance-specific features, whereas **LIMIP still employs a GCNN as its network architecture**, which inherently incorporates instance-dependent feature design. Therefore, it is natural that TreeGate cannot outperform LIMIP in a homogeneous setting. Conversely, LIMIP may also struggle to outperform TreeGate in **truly heterogeneous scenarios**, which aligns perfectly with the reasoning we provided in our earlier rebuttal for not comparing against other baselines. To be more specific, since GCNNs rely on instance-dependent features, our experiments have already demonstrated that **GCNNs perform very poorly on heterogeneous instances**. Therefore, as long as LIMIP continues to use the GCNN architecture (unless we modify the feature design and network structure), comparing it with our approach would not meaningfully evaluate our contribution.
> > >
> > > [1] LIMIP: Lifelong Learning to Solve Mixed Integer Programs, AAAI-2023. (https://arxiv.org/pdf/2208.12226)

---

### Official Review · Reviewer_X7WR · 2025-06-22

**Clarity:** 4
**Significance:** 3
**Originality:** 2
**Rating:** 4
**Confidence:** 4

**Summary:**

This paper presents Mamba-Branching, a novel learning-based branching policy for branch-and-bound (B&B). The approach leverages the Mamba architecture to model the temporal dynamics across B&B iterations and incorporates a contrastive learning framework to pretrain the node embedding layer. Empirical evaluations on real-world MILP datasets show that Mamba-Branching consistently outperforms prior neural baselines and, in some cases, even surpasses the classical solver SCIP.

**Questions:**

1. **Regarding Table 2**, Mamba-Branching explores more nodes than the relpscost heuristic but achieves a smaller primal–dual gap.

- Could this improvement be primarily attributed to **GPU acceleration**?

- Have you evaluated the method’s **performance on less powerful hardware**, such as mid-range GPUs or even CPU-only settings?

- Additionally, could you provide **per-instance comparisons** between Mamba-Branching and relpscost on the **largest few MILP-L instances**?

2. Instead of contrastive pretraining, have you considered **standard supervised pretraining** for the embedding layer?

- For example, by attaching **a linear classification head** to predict the best branching feature, training it to convergence, and then discarding the head.

- How would such a supervised approach compare to contrastive pretraining in terms of **downstream branching performance**?

**Ethical Concerns:**

["NO or VERY MINOR ethics concerns only"]

**Final Justification:**

The rebuttal has effectively addressed part of my concerns, including the advantage on less powerful GPUs and statistical significance against neural baselines. But the concerns in the motivation for contrastive loss and the advantage against the classical solver remains. I would maintain my current score as 4 (borderline accept).

**Reasons for maintaining my score**

The proposed method consistently improves the neural baselines with a significant margin, also with certain advantages against the classical solver. The technical part is also novel.

**Reasons for not a higher score**
1. The motivation of the contrastive loss is not convincing enough to justify why it leads to better performance than standard supervised learning loss. I believe the supervised learning loss itself aims at the two goals outlined by the authors. There lacks a deep understanding in the performance gap.

2. Although Mamba-Branching shows some advantage against relpscost on MILP-L, this margin is not significantly large, as shown in the statistical test results provided by the authors.

**Limitations:**

Yes

**Quality:**

2

**Strengths And Weaknesses:**

**Strengths**

1. To the best of my knowledge, this is the first work to explicitly model step-wise temporal dependencies in B&B, addressing a significant gap in neural branching research.
2. The proposed method demonstrates robust performance on large, heterogeneous MILP instances (MILP-L), where prior neural methods often struggle. In some settings, it even outperforms SCIP.
3. The paper is well-written and well-structured, offering clear explanations of both the methodology and experimental results.


**Weakness**
1. The performance gains over strong baselines, such as TreeGate (on MILP-S) and T-BranT (on MILP-L), are relatively modest. As shown in Figure 2, Mamba-Branching leads in only about 50% of the instances. The paper would benefit from statistical significance testing (e.g., paired t-tests or Wilcoxon signed-rank tests) to validate the robustness of these improvements.
2. There remains a large performance gap in node count compared to the relpscost heuristic. It is also unclear whether the reported advantages on MILP-L are sustained on less powerful hardware than the A100 GPU used.
3. The **motivation for using a contrastive loss** is underdeveloped. While the loss function resembles Deep Graph Infomax (DGI) [2] more than CLIP [1], the paper provides **little justification** for why maximizing local-global mutual information is particularly suitable for node embeddings in B&B.
4. The **connection to established contrastive learning approaches** in NLP or vision (e.g., CLIP) appears weak, especially considering the **supervised nature** of the downstream task.
5. Given this supervision,**simpler pretraining strategies** (e.g., training a linear classifier on node labels and discarding it) should be considered and empirically compared as baselines.


[1] Radford, A., Kim, J.W., Hallacy, C., Ramesh, A., Goh, G., Agarwal, S., Sastry, G., Askell, A., Mishkin, P., Clark, J., Krueger, G., Sutskever, I.: Learning transferable visual models from natural language supervision (2021), https://arxiv.org/abs/2103.00020

[2] Petar Veličković, William Fedus, William L. Hamilton, Pietro Liò, Yoshua Bengio, and R Devon Hjelm. 2018. Deep Graph Infomax. arXiv:1809.10341

---

> ### Author Rebuttal · Authors · 2025-07-30
>
> First of all, we appreciate your recognition of our work, including **addressing a significant gap in neural branching research, robust performance on large, heterogeneous MILP instances, and the paper being well-written and well-structured**. Below, please allow us to respond to all your points regarding weaknesses and questions.
>
> > ### Response to Weakness 1: Statistical Significance Testing
>
> Thank you for your valuable suggestions! We have supplemented detailed statistical analyses in the **"Response to Weakness 1" section for reviewer KMMV**, including significance tests and other statistical results.
>
> For the significance test, we employed the Friedman test with Conover post-hoc analysis, which is a significance test method similar to the Wilcoxon signed-rank test. However, it is **more suitable for our scenario** involving more than two methods.
>
> > ### Response to Weakness 2: Whether It Benefits from Hardware Acceleration
>
> Thank you for your valuable suggestions!  The key concerns you have raised is "Can the results be maintained on devices less powerful than A100 (or even on CPUs)?"
>
> We have provided experimental results on RTX 4090, as shown in the following table:
>
> |             | Mamba-Branching | relpscost |
> |-------------|-----------------|-----------|
> | PD integral | 10351.34        | 10898.59  |
>
> Table 1: Supplementary results obtained on the **RTX 4090** device. The experimental results are the geometric mean of the primal-dual integrals for 16 difficult instances in MILP-L. Compared with previous experiments, no changes were made to the code other than the replacement of hardware equipment. Relpscost was also re-run on the new equipment to ensure a fair comparison.
>
> It can be seen that **Mamba-Branching** still **maintains its advantages** even on GPU that is inferior to the A100. Although the RTX 4090 is still a good GPU, due to constraints on our available equipment, we have been unable to obtain a less powerful GPU for further experiments.
>
> Meanwhile, we attempted to run Mamba-Branching on a CPU, but in the **mamba_ssm** library (the **official code repository** provided by the authors of Mamba), the selective_scan_cuda function requires execution on a GPU and **cannot run in a pure CPU environment**. Therefore, we were unable to test the performance in a pure CPU environment within a short period of time.
> Furthermore, we emphasize that branch policies maintaining efficiency in pure CPU environments are **indeed a widely discussed direction** [1,2,3], but this is **not the focus of our work**.
>
> > ### Response to Weakness 3: Motivation for Using a Contrastive Loss
>
> Regarding the motivation for using the contrastive loss, we have provided a detailed explanation in the **"Response to Question 2" section for Reviewer TcMy**. Due to space constraints, we briefly restate its core idea here: we explain it from **an intuitive perspective** and use **mathematical formulas** to elaborate why local-global mutual information should be maximized.
>
> First, we make the following two assumptions:
>  1. The feature embeddings of the optimal branch decision variable can be distinguished from those of other branch variables in the vector space.
> 2. The feature embedding $e_t$ is informative for branch decision-making.
>
> > #### Intuitive Explanation
>
> Based on assumption 1, we establish an anchor point $e_t^m$ and employ contrastive learning to train the model such that: (1) the optimal branch variable's feature $e_t^a$ is pulled closer to $e_t^m$, while (2) other variables' features are pushed away from $e_t^m$.
> This approach amplifies the separation between optimal and non-optimal features, thereby ensuring sufficient distinguishability when these features are subsequently processed by Mamba.
>
> > #### Mathematical Formulations
>
> For each branching variable $i$, we define $F(i)$ as the branching score function for variable $i$, such that the optimal branching variable satisfies $$a = \arg\max_{i} F(i).$$
>
> Assumption 2 posits that the embeddings can encode branching quality information, which we simplify to a linear relationship $$e_t^i = \alpha F(i) + \epsilon_i,$$where $\alpha \in \mathbb{R}^d$ is a variable independent vector. $\epsilon_i \in \mathbb{R}^d$​ is a noise term satisfying $\mathbb{E}(\epsilon_i)=0$ with $|\epsilon_i| \approx 0$.
>
> We use max pooling over the embeddings as the anchor:
> $$e_t^m
> =[...,\max_i (F(i)\alpha(j)+\epsilon_i(j)),...].$$
> For $e_t^a$, we have:
> $$e_t^a=[...,\max_i F(i)\alpha(j) + \epsilon_a(j),...].$$
> Indeed, we can derive the approximation $e_t^a \approx e_t^m$, then reasonably establish that:
> $$\forall i \neq a,\text{sim}(e_t^a, e_t^m) > \text{sim}(e_t^i, e_t^m).$$
> This formulation is mathematically equivalent to the expression presented in the original paper.
>
> > ### Response to Weakness 4: Connection to Established Contrastive Learning Methods
>
> We acknowledge that our research is indeed not strongly related to CLIP and is actually more similar to DGI. However, rather than a strict extension of CLIP, we merely drew **inspiration** from contrastive learning methods like CLIP. We will add DGI to the introduction (alongside CLIP) in the revised paper to highlight these as our inspirational sources.
>
> > ### Response to Weakness 5 and Question 2: A Simpler Pretraining Strategies
>
> Thank you for your suggestions. You have indeed proposed an idea that is more intuitive, natural, and simpler compared to contrastive learning: _attaching a linear classification head to predict the best branching feature, training it until convergence, and then discarding the head_.
>
> Unfortunately, in our work, we adopted this approach **from the very beginning** but found that **its performance was unsatisfactory**. Subsequently, we further reflected and attributed the poor performance to the **insufficient discriminative power** of the embedding features of each variable. Therefore, we further designed a contrastive learning method to address this issue.
>
> We are happy to provide the results of the training conducted using your proposed approach, named **Mamba-Branching-classifier**.
> Specifically, we first trained a TreeGate network, discarded its linear classification head (retaining only the embedding feature output), and then trained it jointly with Mamba.
>
> |            | Mamba-Branching | Mamba-Branching-classifier | TreeGate |
> |------------|-----------------|----------------------------|----------|
> | Nodes      | 2054.99         | 3353.60                    | 2171.31  |
> | Fair Nodes | 2077.55         | 3420.68                    | 2205.06  |
>
> Table 2: Validation of the approach "training a linear classifier on labels and discarding it", named Mamba-Branching-classifier. The experiments are conducted on MILP-S. Compared to Mamba-Branching, Mamba-Branching-classifier **merely** replaces the contrastive loss with the aforementioned method.
>
> From this set of results, Mamba-Branching significantly outperforms Mamba-Branching-classifier, and the latter fails to surpass TreeGate, which demonstrates the **importance of contrastive learning**.
>
> > ### Response to Question 1: Performance on Less Powerful Hardware and Per-Instance Comparisons
>
> > #### Performance on Less Powerful Hardware
>
> We fully understand your concerns and have provided results on less powerful hardware in **"Response to Weakness 2"**.
> It is important to emphasize that our method benefits from GPU acceleration but is **not entirely attributed to** it. We believe that this series of machine learning-based methods all attributed to **data-driven characteristics**; thus, [1, 2, 3] achieve high performance on CPUs, yet this is **not the focus of our work**.
>
> > #### Per-instance comparison
>
> We are pleased to provide the results for each hard instance in MILP-L as follows:
>
> |                          | Mamba-Branching | relpscost     |
> | ------------------------ | --------------- | ------------- |
> | atlanta-ip               | 24258.31        | **24170.83**  |
> | bab2                     | **38080.78**    | 42185.32      |
> | bab5                     | 3294.13         | **2881.17**   |
> | bab6                     | **20686.83**    | 22492.31      |
> | harp2                    | **44.92**       | 76.33         |
> | map16715-04              | 65686.53        | **59963.96**  |
> | msc98-ip                 | **1322.72**     | 1530.67       |
> | mspp16                   | 37661.03        | **35465.03**  |
> | n3seq24                  | **7145.50**     | 7836.84       |
> | neos-4338804-snowy       | **5880.37**     | 5880.46       |
> | neos-4387871-tavua       | **67630.53**    | 72370.35      |
> | neos-4647030-tutaki      | **2463.27**     | 2484.55       |
> | nursesched-medium-hint03 | 115857.37       | **114998.07** |
> | opm2-z10-s4              | **107706.76**   | 112561.74     |
> | pigeon-10                | 36003.53        | **36003.52**  |
> | radiationm40-10-02       | **597.08**      | 624.90        |
>
> Table 3: Comparison of primal-dual integral between Mamba-Branching and relpscost on 16 hard instances in MILP-L. For each instance, the geometric mean of the results across 5 seeds is calculated.
>
> Of **16** instances, Mamba-Branching leads in **10** compared to relpscost.
> This is similar to the win/tie/loss table in **"Response to Weakness 1" by Reviewer KMMV**, except that the median of each instance is replaced with the geometric mean, further enriching the experimental results.
>
> > ### Reference
>
> [1] Gupta, Prateek, et al. "Hybrid models for learning to branch." _Advances in neural information processing systems_ 33 (2020): 18087-18097.
>
> [2] Kuang, Yufei, et al. "Rethinking branching on exact combinatorial optimization solver: The first deep symbolic discovery framework." _The Twelfth International Conference on Learning Representations_. 2024.
>
> [3] Kuang, Yufei, et al. "Towards general algorithm discovery for combinatorial optimization: Learning symbolic branching policy from bipartite graph." _Forty-first International Conference on Machine Learning_. 2024.

---

> > ### Comment · Reviewer_X7WR · 2025-08-05
> >
> > I highly appreciate the detailed response and additional results, which have effectively addressed part of my concerns, including the advantage on less powerful GPUs and statistical significance against neural baselines. I decide to **maintain my current score** for the following reasons.
> >
> > **Reasons for maintaining my score**
> >
> > The proposed method consistently improves the neural baselines with a significant margin, also with certain advantages against the classical solver. The technical part is also novel.
> >
> > **Reasons for not a higher score**
> > 1. The motivation of the contrastive loss is not convincing enough to justify why it leads to better performance than standard supervised learning loss. I believe the supervised learning loss itself aims at the two goals outlined by the authors. There lacks a deep understanding in the performance gap.
> >
> > 2. Although Mamba-Branching shows some advantage against relpscost on MILP-L, this margin is not significantly large, as shown in the statistical test results provided by the authors.

---

> ### Author Response · Authors · 2025-08-05
> **Acknowledgment to the Reviewer and Supplemental Remarks**
>
> Thank you very much for your **valuable suggestions** and the **effort** you have put into reviewing our paper! The questions you raised have provided us with **excellent insights** and have helped make our work more comprehensive. Regarding the two remaining points of confusion, we fully understand your perspective, and we would like to supplement the following:
>
> > ### Response to Q1: Motivation of the Contrastive Loss
>
> First of all, we fully acknowledge that you are absolutely right: indeed, the standard supervised learning loss is also intuitively reasonable. We admit that we cannot provide a rigorous mathematical proof through more detailed theoretical analysis to demonstrate the advantage of contrastive loss over supervised learning loss.
>
> However, our **experimental results** validate that **contrastive loss achieves better performance than supervised learning loss** -- in fact, the method based on **supervised learning loss even fail to surpass the baselines**.
>
> Your suggestion is truly insightful: a more detailed theoretical analysis behind these experimental results presents a challenging yet promising new direction. For now, what we can do is to **demonstrate the superiority of contrastive loss through experimental evidence**.
>
> > ### Response to Q2: Performance Improvement is not Statistically Significant
>
> We fully agree with your observation that while the statistical results demonstrate Mamba-Branching's superiority over relpscost, the advantage does not reach statistical significance. We would like to highlight that our method **represents meaningful progress** compared to previous baselines [1,2] -- whereas existing baselines fail to outperform relpscost in any scenario, our approach achieves superior performance on the most practically relevant and challenging real-world problems.
>
> Your expectation is indeed insightful -- achieving statistically significant improvements over relpscost remains an important goal for future work in this field. At present, **Mamba-Branching**'s demonstrated capability is to **significantly outperform other baselines** while also **surpassing relpscost**.
>
> > ### Acknowledgement
>
> We sincerely appreciate your valuable contributions to our work! Your insightful comments have provided us with unique perspectives that have significantly helped improve our paper.
>
> > ### Reference
>
> [1] Zarpellon, Giulia, et al. "Parameterizing branch-and-bound search trees to learn branching policies." Proceedings of the aaai conference on artificial intelligence. Vol. 35. No. 5. 2021.
>
> [2] Lin, Jiacheng, et al. "Learning to branch with Tree-aware Branching Transformers." Knowledge-Based Systems 252 (2022): 109455.

---

> > ### Comment · Reviewer_X7WR · 2025-08-06
> >
> > Thank you again for the followup.
> >
> > >  demonstrate the superiority of contrastive loss through experimental evidence
> >
> > Although I highly appreciate the empirical discovery here, I would like to point out that to fully validate this contrastive loss, you need a more comprehensive analysis (not necessarily theoretical analysis), such as
> > - comparing different contrastive learning and supervised learning strategies,
> > - including more datasets,
> > - applying this off-the-shelf contrastive learning stage to other neural branching methods to show its universal application.
> >
> > But I believe performing the above analysis is out of the scope of this work since this it mainly focuses on capturing the long-term dependency in branching. My personal suggestion is to not highlight this part too much (even removing it is fine) but just treat it as a small trick, otherwise it would hurt the main contribution from your Mamba part. More rigorously, you should also apply this trick to other neural baselines for comparison since it is independent of the Mamba method.

---

> > > ### Author Response · Authors · 2025-08-07
> > >
> > > Thank you for your valuable suggestions! Your comments have provided us with **significant inspiration**. Indeed, as you pointed out, the **contrastive learning method in our paper serves sequence modeling**. Furthermore, your suggestion about potentially integrating contrastive learning with branching policies in a broader context is truly insightful. Currently, contrastive learning is only applied to sequence modeling (as demonstrated in our original experiments and supplementary rebuttal experiments for **Mamba, Transformer, and GRU**). However, extending contrastive learning to a **wider** range of branching policies represents a highly promising new research direction. We find your suggestion both inspiring and forward-looking.
> > >
> > > Additionally, if we may, we have obtained **a new set of experimental results** and would like to further clarify why Mamba-Branching outperforms relpscost primarily in complex heterogeneous scenarios -- this is largely due to the **intentionally high difficulty of our experimental setup**. Our research focuses on heterogeneous scenarios, which are **inherently challenging**. However, when we reduce the experimental difficulty (e.g., in **homogeneous** scenarios), our latest results show that **Mamba-Branching can indeed surpass relpscost**.
> > >
> > > Please allow me to quote our response to Reviewer 3WGQ as follows:
> > >
> > > > #### Results on homogeneous problems
> > >
> > > Many reviewers have raised concerns that imitating relpscost may not surpass relpscost itself. Therefore, here we conduct experiments on **homogeneous** scenarios. Specifically, we use **Set Covering** for both training and testing instances, only **varying the instance parameters**. The results are as follows:
> > >
> > >
> > > | Method | Mamba-Branching | TreeGate | Relpscost |
> > > |--------|-----------------|----------|-----------|
> > > | Nodes  | 265.07      | 280.38   | 170.81    |
> > > | Time   | **9.64**        | 9.89     | 11.96     |
> > >
> > > Table 1:  Results of training and testing on **Set Covering**. Both Mamba-Branching and TreeGate were initialized with relpscost once. During training, the expert policy being imitated was **relpscost**, as the previous section has already demonstrated that **imitating Strong Branching leads to significantly worse training accuracy**.
> > >
> > > This result clearly demonstrates one key point: **imitating relpscost does not inherently prevent a branching policy from surpassing relpscost**. However, why does Mamba-Branching only exceed relpscost on complex problems in heterogeneous scenarios? We argue that this is fundamentally due to the **inherent complexity of heterogeneous settings** -- all neural networks face generalization challenges, whereas relpscost remains consistent across any problem. **The intrinsic difficulty of heterogeneous scenarios leads to Mamba-Branching surpassing relpscost only in complex instances**. However, if the problem difficulty is reduced (e.g., by switching to homogeneous settings), exceeding relpscost is not impossible.
> > >
> > > Therefore, we would like to emphasize: **Mamba-Branching can surpass relpscost in complex heterogeneous scenarios, representing an improvement over prior work**. However, **the concern that Mamba-Branching does not consistently outperform relpscost stems from our deliberate choice of a challenging setting** -- heterogeneous scenarios. **In homogeneous problems, Mamba-Branching does achieve superior performance over relpscost**.

---

### Official Review · Reviewer_QH4H · 2025-06-23

**Clarity:** 4
**Significance:** 2
**Originality:** 3
**Rating:** 3
**Confidence:** 5

**Summary:**

The authors propose Mamba-Branching, a method for enhancing heterogeneous MILP solving. Compared to Transformer- and GCNN-based baselines, Mamba-Branching demonstrates higher efficiency and stronger capability in capturing temporal dynamics across branch-and-bound (B&B) steps. The introduction of contrastive pretraining further boosts its performance. Experimental results show that Mamba-Branching excels particularly on hard instances—those unsolved within a 3600-second time limit—achieving the lowest primal-dual integral (PD Integral), even outperforming the expert policy used for imitation learning. However, on easier instances, its improvements are marginal over the baseline TreeGate, and it does not surpass the expert policy, relpcost.

**Questions:**

Q1. Could the authors provide evaluation based on solve time, especially for easy MILP instances, as discussed in the Weakness section? From the code and result logs, it appears that Mamba-Branching is actually slower than the TreeGate baseline and probably much slower than relpcost expert stratery, suggesting no clear advantage in actual solver efficiency. The calculation could be 1-shifted geometric mean of the solving times in seconds, following https://arxiv.org/pdf/1906.01629.


Q2. I understand that heterogeneous MILP solving with machine learning is a relatively challenging task. In fact, none of the recent works—including TreeGate, T-BranT, and the current Mamba-Branching method—have been able to outperform the expert branching policy relpcost. This is problematic because the whole idea behind imitation learning is to either:
* (1) Approximate a powerful but slow expert policy in order to accelerate solving; for instance, in the context of ML for B&B, the GCNN paper (https://arxiv.org/pdf/1906.01629) uses strong branching—a very costly but high-quality decision policy—as the expert. They then train a neural network to imitate strong branching, achieving faster solving on homogeneous MILPs.
* (2) Use imitation learning as a pretraining stage for reinforcement learning, with the goal of surpassing the expert policy via fine-tuning.

However, TreeGate, T-BranT, and Mamba-Branching fail to realize either of these benefits. On easy MILP instances, their performance in terms of both node count and solve time is still worse than relpcost. But if the learned policy doesn't outperform the expert in any meaningful way, then just using relpcost directly is better.

Q3. To be fair, this paper offers meaningful technical innovations. If the authors are interested in strengthening the work further or move forward to the next paper, I suggest two possible directions:

(1) Apply Mamba to homogeneous MILP solving, following the setup in https://arxiv.org/pdf/1906.01629. That paper shows that a learned policy can outperform traditional methods—including strong branching and relpcost—on time efficiency. Your model could potentially improve decision quality due to Mamba’s ability to model temporal structure. The key is to balance the computational cost of Mamba with the gain in decision quality.

(2) If the focus remains on heterogeneous MILP solving (as in this paper), I strongly encourage the authors to carefully revisit the hand-crafted features used in TreeGate. These features may not be well-suited for the type of expert policy being imitated. Specifically, relpcost is not a single coherent policy but a mixture of two strategies, i.e.,  strong branching and pseudo-cost branching. If you examine relpcost in detail, you'll find it blends strong and pseudo decisions. However, features extracted in TreeGate—such as SCIPgetAvgPseudocostScore in the average score vector—are biased toward pseudo branching. As a result, the learning model may simply overfit to pseudo-branch decisions because they are easier to imitate given the input representation. This partially explains why both TreeGate and T-BranT report high imitation accuracy, but not necessarily meaningful performance gains in solving. The expert actions the models are imitating are pseudo branching which is of lower quality than strong branching.

Therefore, instead of relying on TreeGate’s features and mimicking relpcost, I suggest exploring the GCNN-based features and strong branching as the expert. Admittedly, this is difficult. As you have also shown empirically, GCNN + strong branching, which works well in homogeneous settings, can be hard to transfer to heterogeneous ones. This might be due to the limited expressiveness of GCNNs in capturing features across MILPs. Hence, future work could focus on strengthening the model architecture and leveraging large-scale pretraining to improve generalization.

**Final remark:** Overall, this is a well-written paper with thoughtful modeling and clear motivation. However, because it follows the evaluation setup and assumptions of prior works that may not be rigorous—particularly in metric selection, input features and imitation target choice—I believe it falls short for acceptance at NeurIPS. The paper may be better suited for other venues.

**Ethical Concerns:**

["NO or VERY MINOR ethics concerns only"]

**Final Justification:**

The authors spent enough efforts for the rebuttal, but some key issues in the ML for heterogeneous MILP solving domain has not been solved. But the architecture and neural networks proposed in this paper is still good, but don't contribute too much for this domain. Therefore, I raised my score from 2 to 3.

**Limitations:**

See the Questions.

**Quality:**

3

**Strengths And Weaknesses:**

Strengths:
1.  The paper is clearly structured and well-articulated, making it easy to follow the motivation, methodology, and results. The presentation effectively guides the reader through complex technical details without sacrificing clarity.
2. The application of Mamba to model the trajectory of branch-and-bound (B&B) decisions is both novel and insightful. By capturing the temporal evolution of the search tree, the method leverages sequential dependencies across B&B steps to enhance branching decisions, demonstrating a deep understanding of the problem structure.

Weaknesses:
1. Inappropriate evaluation metric for easy instances: The paper uses the number of nodes explored as the evaluation metric for easy MILP instances, justifying it by stating that it "directly impacts overall solving time. (https://arxiv.org/pdf/1906.01629)" However, a more appropriate and direct metric for easy instances is the solve time itself. In such settings, node count may not reliably reflect actual solver efficiency.
2. No advantage in solve time over baselines: From the provided code and result logs, it is evident that Mamba-Branching does not outperform the TreeGate baseline in terms of actual solving time. In fact, Mamba-Branching often requires longer wall-clock time, suggesting that its more complex modeling of B&B trajectories, while conceptually interesting, does not translate into practical speedup—especially in easy instances where fast convergence is crucial.

---

> ### Author Rebuttal · Authors · 2025-07-25
>
> First, we appreciate your recognition of the strengths of our work, including **the clarity of the paper writing, the well-defined motivation, the novel and insightful Mamba sequence modeling, and the effective contrastive learning pre-training**, as well as your acknowledgment of the **high efficiency in solving difficult problems**.
> Next, let me address your points regarding weaknesses and the questions raised.
>
> > ### Response to Weaknesses 1 and 2
>
> > #### Weakness 1: Evaluation metrics for easy instances, nodes or time?
>
> First, we sincerely thank you for valuable feedback.
> In this field that emphasizes the generalization of branching policies, the **two prior works [1, 2]** evaluated easy instances **using only nodes and fair nodes as metrics**.
> This is because solving time is influenced not only by the node count (fewer nodes generally lead to fewer simplex iterations) but also by the neural network’s inference time, which can vary significantly across hardware devices. Thus, while we included **solving time** in our experiments **for reference**, we emphasize that **a completely fair comparison** is **challenging** due to potential **discrepancies in hardware** environments across methods.
>
> > #### Weaknesses 2: No advantage in solve time over baselines.
>
> However, we are happy to provide the solving time results as follows.
>
> | Method            | MILP-S | MILP-L easy |
> | ----------------- | ------ | ----------- |
> | TreeGate-p        | 163.39 | \           |
> | TreeGate          | 111.17 | 145.44      |
> | Mamba-Branching-p | 147.96 | \           |
> | Mamba-Branching   | 112.32 | **123.79**  |
> | T-BranT           | 165.59 | 210.27      |
>
> table 1: The solving time results for several competitive methods on **MILP-S (8 instances)** and easy instances in **MILP-L (57 instances)**. Here, "-p" indicates the pure use of neural networks. It should be noted that, except for the results of TreeGate and T-BranT under MILP-L, which are supplemented in the table here, all other results **can be found** in the result logs.
>
> We acknowledge that among the three MILP-S groups initialized with a single relpscost (TreeGate, T-BranT, Mamba-Branching), TreeGate has a slight advantage (111.17 vs. Mamba-Branching's 112.32).
> However, after **excluding relpscost initialization** (TreeGate-p vs. Mamba-Branching-p), **Mamba-Branching-p (147.96) is faster than TreeGate-p (163.39)**.
> Furthermore, for the easy instances **in MILP-L, Mamba-Branching (123.79) achieves the fastest solving time** among all neural branching policies.
> Compared with the **8 instances in MILP-S**, the **57 instances in MILP-L** are more **convincing**.
>
> The reasons for these observations can be summarized as follows:
>
> - **Architectural Complexity**: Mamba-Branching's inherently more complex network structure leads to longer inference times, explaining why TreeGate can sometimes achieve shorter solving times despite exploring more nodes.
> - **Scalability Advantage**: For MILP-L with **more test instances (57 vs. 8)**, Mamba-Branching's **superior node efficiency** offsets its higher computational overhead, resulting in **better overall solving performance**.
> - **Initialization Impact**: The impact of initialization may also lead to unfair comparison, as factors other than the performance of the neural network are involved. After **excluding relpscost initialization, Mamba-Branching-p consistently demonstrates faster solving times**.
>
> In summary, the statement that _"No advantage in solve time over baselines, Mamba-Branching does not outperform the TreeGate baseline in terms of actual solving time"_ requires **clarification**: While TreeGate is marginally faster on one specific MILP-S (8 test instances), **Mamba-Branching demonstrates significantly better performance on MILP-L easy instances (57 test instances)**, which better represents its practical effectiveness.
>
> > ### Response to Q1: Evaluation Based on Solving Time
>
> In our responses to Weakness 1 and Weakness 2, we have provided solving time results and corresponding analyses. The results demonstrate that while **Mamba-Branching** performs slightly worse than TreeGate on MILP-S, it **outperforms TreeGate on more test instances (easy instances of MILP-L)** and **surpasses TreeGate in pure neural network tests** where the influence of relpscost is excluded. The reasons for these observations have been thoroughly analyzed in the aforementioned responses.
>
> We would like to emphasize that all neural branching policies are indeed slower than relpscost due to the substantial gap in node counts on easy instances -- this is an undeniable fact.
> However, it should be noted that **neither TreeGate nor T-BranT surpass relpscost** in **any** scenario, whereas **Mamba-Branching** achieves **superior performance** over relpscost on **hard instances** of MILP-L.
> Furthermore, Section 5.2.3 of our original paper provides a detailed analysis of the underlying reasons for this phenomenon. In fact, the strong branching initialization of **relpscost** is particularly time-consuming for complex problems, which makes it **lose its advantages in difficult scenarios** and perform **worse than Mamba-Branching**.
> A more comprehensive comparison and analysis between neural branching policies and relpscost will be presented in our response to Q2.
>
> > ### Response to Q2: Machine Learning Cannot Surpass Expert Strategy Relpcost
>
> I fully understand your concern regarding TreeGate, T-BranT, and Mamba-Branching not surpassing relpscost on easy instances. Let me address this from two perspectives: first, the **original explanations provided by TreeGate and T-BranT** regarding this matter, and second, the **advantages of our Mamba-Branching compared to relpscost** as presented in our work.
>
> > #### Original Explanations of TreeGate and T-BranT
>
> - TreeGate's original paper states _"While we train our BVS policies to imitate the SCIP default policy relpscost, our objective is not to outperform relpscost. Indeed, expert BVS policies like relpscost are tuned over thousands of solvers' proprietary instances: comprehensively improving on them is a very hard task, impossible to guarantee in our purely-research experimental setting, and should thus not be the only yardstick to determine the validity (and practicality) of a learned policy."_
> - Similarly, T-BranT explains _"In fact, as implemented in the SCIP solver, branching with relpscost will trigger side-effects that modify the internal SCIP states and facilitate the following branching processes. And it is hard to re-implement a pure version of relpscost with its peculiarities retained. Thus, we do not judge our models by whether they can surpass relpscost as Zarpellon et al. [1] do."_
>
> As evident, both papers provide clear explanations for why **direct comparison with relpscost may not be necessary or appropriate**.
>
> > #### Advantages of Mamba-Branching Compared to relpscost
>
> Compared to TreeGate and T-BranT, **Mamba-Branching** has **already demonstrated the potential to surpass relpscost**, outperforming it on difficult problems in MILP-L. In my opinion, the fundamental reason lies in the fact that complex problems involve a larger number of variables, making the strong branching initialization process of relpscost excessively time-consuming. As a result, **relpscost loses its advantage on large-scale problems**. While Mamba-Branching may not perform as well as relpscost on simpler problems, its **superior capability** on more **realistic and practically valuable complex** problems should **not be overlooked**.
>
> > ### Response to Q3: Suggestions for Strengthening the Work
>
> We sincerely appreciate your valuable suggestions, which reflect your profound understanding of the field. As you aptly noted, GCNN shows poor generalization on heterogeneous problems -- this is precisely what motivated our adoption of the TreeGate feature + relpscost. Regarding your comments on the **TreeGate feature + relpscost expert**, we would like to elaborate on the **advantages** of this paradigm over **GCNN + strong branching**.
>
> 1. As noted in [1], relpscost represents a more practical expert since strong branching is rarely used directly in real-world problem solving. This provides significant benefits during **dataset collection** -- the time required for gathering relpscost data is substantially less than for strong branching. While we cannot provide exact quantitative comparisons (as this wasn't our paper's focus), empirical evidence shows **strong branching becomes prohibitively slow** for complex problems, whereas **relpscost remains computationally efficient**.
> 2. As you mentioned, TreeGate effectively addresses the generalization issue. They emphasize incorporating **tree-search-related features** into the design without focusing on specific instances, while **pseudo-costs align with the requirements of tree search** by leveraging historical information. Therefore, they integrate a substantial number of pseudo-costs into the features (SCIPgetAvgPseudocostScore).
> 3. Relpscost essentially represents an improved form of pseudo-cost that incorporates historical strong branching experience. The features capture **"imperfect" pseudo-costs** while the relpscost labels contain **"refined" pseudo-costs**. The neural network learns to transform pseudo-costs **from imperfect to refined** versions -- this represents meaningful learning rather than overfitting as suggested.
>
> In summary, we believe the TreeGate feature design combined with relpscost expert represents **a necessary choice to achieve generalization capability, while maintaining dataset collection efficiency.**
>
> > ### Reference
>
> [1] Zarpellon, Giulia, et al. "Parameterizing branch-and-bound search trees to learn branching policies." Proceedings of the aaai conference on artificial intelligence. Vol. 35. No. 5. 2021.
>
> [2] Lin, Jiacheng, et al. "Learning to branch with Tree-aware Branching Transformers." Knowledge-Based Systems 252 (2022): 109455.

---

> > ### Comment · Reviewer_QH4H · 2025-08-02
> > **Response to the Authors**
> >
> > Thank you for your rebuttal and the effort you've put into addressing the review. However, I believe this paper—along with the TreeGate line of work it builds upon—still has several fundamental issues. Below, I respond to your key points:
> >
> > (1) regarding the explanation given in TreeGate and repeated in T-BranT:
> >
> > T-BranT essentially repeats TreeGate’s explanation, but the explanation in TreeGate itself is rather vague. It claims that relpscost is difficult to tune and replicate, but this is not well supported—in fact, relpscost is not particularly parameter-sensitive. Moreover, relpscost is not an unbeatable policy. Again, the GCNN paper [https://arxiv.org/pdf/1906.01629] demonstrates that relpscost is outperformed by GCNN in homogeneous settings. This leads to a crucial point: solve time is the most important metric, especially in real-world applications where the goal is to find solutions quickly and efficiently.
> >
> > (2) I would suggest a simple but insightful experiment that could provide clarity on what your neural policy is actually learning. In your collected dataset from relpcost, you could divide the samples into two groups—those generated via strong branching and those via pseudo-cost branching. Then, conduct an ablation study: train separate imitation learning models using only the strong-branching decisions and only the pseudo-cost ones. Report the imitation accuracy and solving time for each. If time is limited, running this only on the easy instances would still be valuable. This could provide a clearer picture of whether your model is overfitting to pseudo-cost or actually learning meaningful decision-making.
> >
> > (3) regarding your argument:
> > > “As noted in [1], relpscost represents a more practical expert since strong branching is rarely used directly in real-world problem solving...”
> >
> > This is not a sufficient reason to justify relpscost as a good expert for imitation learning. As I mentioned earlier in my review, the core idea behind imitation learning is to either:
> > 1. Approximate a powerful but slow expert (e.g., strong branching) in order to achieve faster solving time with a learned policy. This is what the GCNN paper does—they use strong branching as the expert, then train a neural network that imitates it efficiently.
> > 2. Use imitation learning as a pretraining step for reinforcement learning, with the goal of surpassing the expert policy through further optimization.
> >
> > So, prioritizing an expert policy merely because it's easier or cheaper to collect data from (as in the case of relpscost) is not ideal. The goal should be to imitate a high-quality expert, even if collecting data from it is costly. This allows the neural network to gradually learn and accelerate strong-branching-type decision making through approximation—thus achieving both quality and efficiency.
> >
> > (4)  regarding your explanation:
> > > “...solving time is influenced not only by the node count (fewer nodes generally lead to fewer simplex iterations) but also by the neural network’s inference time, which can vary significantly across hardware devices...”
> >
> > This reasoning is not entirely convincing. While it is true that inference time may differ across hardware setups, a fair time-based evaluation is still very much feasible—as long as all experiments are run on the same server, and CPU/GPU utilization is properly controlled. This is standard practice in experimental ML research, and such variance can be minimized to a negligible level with careful experimental design.
> >
> > Moreover, the claim that “fewer nodes generally lead to shorter solving time” is not necessarily accurate. If you refer to Table 2 in https://arxiv.org/pdf/1906.01629, you’ll see clear cases where methods that explore fewer nodes actually take longer to solve the instance [Maximum Independent Set, Hard, Relpcost/RPB vs. GCNN].
> >
> > Thus, wall-clock solve time should remain the primary evaluation metric, especially in practical applications where total time-to-solution is the most relevant indicator of solver performance.

---

> ### Author Response · Authors · 2025-08-03
> **Response to 1,2,3: Strong Branching or Relpscost? Further Comparative Experiments**
>
> We sincerely appreciate your feedback! We are more than happy to continue discussing the issues you raised -- although we would like to first clarify that your questions are not actually directed at our work, but rather challenge certain aspects of previous research in this field[1,2] -- nevertheless, we are very willing to address your questions.
>
> > ### Response to 1,2,3: Strong Branching or Relpscost? Further Comparative Experiments
>
> We thank you for raising this insightful experiment! To **maintain the generalization capability of the branching policy** (this is a fundamental premise of our work, as we aim to focus specifically on **generalization** performance), we adopted the **same feature design and network architecture as TreeGate [1]**. Under this framework, we trained policies using both **strong branching** and **relpscost** as experts, with results below:
>
> |                             | Train Loss | Test Loss | Train Accuracy | Test Accuracy | Solving Time |
> | --------------------------- | ---------- | --------- | -------------- | ------------- | ------------ |
> | TreeGate + Relpscost        | 0.9954     | 0.9907    | 0.7453         | 0.7561        | 137.41       |
> | TreeGate + Strong Branching | 2.1845     | 2.5912    | 0.3590         | 0.2465        | 512.32       |
>
> Table 1: The imitation learning results using **TreeGate** as the network framework, with **strong branching** and **relpscost** serving as the experts respectively. The training was conducted for 50 epochs in all cases. The table reports the final network's loss and accuracy on both the training and test sets, as well as the solving time performance on test instances from MILP-S.
>
> The results demonstrate that under the TreeGate framework, **imitating strong branching achieves significantly lower accuracy compared to imitating relpscost**, making training considerably more difficult. Meanwhile, on test instances, the **policy imitating relpscost shows clearly superior performance**, having learned **meaningful decision-making** patterns.
>
> As you mentioned, the purpose of imitation learning is:
> _"To approximate a powerful but slow expert (e.g., strong branching) in order to achieve faster solving time with a learned policy. This is what the GCNN paper does—they use strong branching as the expert, then train a neural network that imitates it efficiently."_
>
> We agree that strong branching is a powerful expert in contexts like GCNN. However, our experiments under TreeGate's feature design reveal that relpscost is **a more suitable expert** for learning **generalizable policies**, likely due to **TreeGate's generalization-oriented feature design, which exhibits better compatibility and alignment with relpscost**.
>
> Therefore, under the current feature design, imitating **relpscost** demonstrates more effective learning compared to imitating **strong branching**. While modifying the feature design or network architecture (e.g., abandoning TreeGate’s framework) could be an alternative approach, we note that:
>
> 1. **Our paper’s primary contribution lies in Mamba-based sequential modeling and contrastive learning**, rather than in re-evaluating the feature design choices of prior methods. Indeed, if we abandon TreeGate's generalization-oriented feature design, how to develop features that are **both compatible with strong branching and maintain generalization capability** presents a **highly challenging new research direction**. We must acknowledge your **remarkable insightfulness** -- this new research direction is indeed what you anticipated to see. However, unfortunately, **this lies entirely outside the scope of our current research focus**.
> 2. These comparative experiments are intended to **analyze TreeGate’s behavior under different experts**, rather than to serve as an ablation study for our proposed method.
>
> We appreciate the reviewer’s perspective and are happy to further discuss these trade-offs if needed.
>
> > ### Reference
>
> [1] Zarpellon, Giulia, et al. "Parameterizing branch-and-bound search trees to learn branching policies." Proceedings of the aaai conference on artificial intelligence. Vol. 35. No. 5. 2021.
>
> [2] Lin, Jiacheng, et al. "Learning to branch with Tree-aware Branching Transformers." Knowledge-Based Systems 252 (2022): 109455.

---

> > ### Comment · Reviewer_QH4H · 2025-08-05
> > **Response to the Authors**
> >
> > Thank you for your response and for all the effort you’ve put into this work. I’m sorry, but I still feel that this paper does not represent a major breakthrough. Compared to other fields where publishing is relatively easier, this area might seem a bit harsh—but MILP optimization is inherently more complex than some other domains.
> >
> > You claim that the main contributions lie in the Mamba-based sequential modeling and contrastive learning. However, from my experience working in the ML/RL for combinatorial optimization field for quite some time, simply proposing a new neural network or method is not in itself a significant contribution to this domain.
> >
> > For this paper, the core problem that needs to be addressed is how to effectively model or extract features so that a lightweight neural network can learn high-quality branching policies. However, as it stands, your proposed method—as well as previous related works—still doesn't outperform the expert policy, i.e., relpcost. If that’s the case, then why not just use relpcost directly? Neural networks also require GPU resources to run, and not every practitioner in the optimization community has access to such hardware.
> >
> > Therefore, I believe the key to making real progress lies in revisiting the feature extraction process and ensuring that it’s done correctly. That’s the only way to fundamentally address the problem. While your current work does have its merits, I don’t believe the contributions are sufficient for a NeurIPS publication.
> >
> > But whatever, I have raised my score from 2 to 3, still around and below the borderline. Thanks for the authors' efforts.

---

> ### Author Response · Authors · 2025-08-03
> **Response to 4: Reiteration of the Relationship Between Node Count and Solving Time**
>
> We appreciate the opportunity to clarify this important point. As previously mentioned in our response, we have **consistently** maintained that solving time is determined by **both the number of nodes explored and the inference time of neural networks (or other)**. As noted in our original submission:
>
> _"The solving time is influenced not only by node count (where fewer nodes typically reduce simplex iterations) but also by the neural network's inference time."_
>
> To be precise, our claim means fewer nodes generally lead to **reduced simplex iterations**, rather than asserting that the number of nodes is the sole determinant of solving time. This distinction is crucial.
>
> The observed phenomenon that **relpscost explores fewer nodes than GCNN yet requires longer solution time** can be attributed to the **computational overhead of its strong branching in early stages**. This finding is **consistent** with our explanation: the strong branching employed by relpscost in early stages leads to **prolonged inference time**. Although it results in fewer nodes explored (and consequently fewer simplex iterations), the **disadvantage in inference time** ultimately causes relpscost to be slower than GCNN in overall solution time. This serves as compelling evidence of how these two factors jointly influence the solving time.
>
> Moreover, for the same reason, the strong branching initialization of relpscost proves excessively time-consuming on **complex problems**, which explains **why Mamba-Branching outperforms relpscost in such scenarios**.
>
> And as you requested, we have carefully conducted additional experiments evaluating solving time performance. The results **consistently** demonstrate Mamba-Branching's advantages over baseline methods, validating our contributions.

---

> ### Author Response · Authors · 2025-08-05
> **Response to Reviewer’s Final Decision -- Summary and Acknowledgments**
>
> Thank you for your response! The discussion has been highly beneficial to me. It is evident that you possess profound understanding and unique insights in this field, and I truly cherish this valuable opportunity to exchange ideas with you.
>
> I fully acknowledge your perspective on our work: it is undeniable that we proposed a new neural network/method. Compared to those works that have made exceptionally outstanding contributions, we must admit that our contribution is relatively modest. However, please allow me to provide some additional clarification regarding our work.
>
> > ### Several Pioneering Works
>
> First, this field originated from two seminal papers [1,2]. Both studies designed key features based on critical information in the branch-and-bound and employed machine learning/deep learning to address the variable selection problem. As you rightly pointed out, these works made **highly pioneering contributions** to the field -- they required not only an **in-depth understanding of MILP solvers** to design those crucial features but also **a profound grasp of neural network techniques** to develop specialized architectures.
>
> Following the same reason, I personally believe that TreeGate [3] is also a **pioneering** work. They proposed a generalizable branching policy from two perspectives: **MILP-knowledge-based feature design** and **new neural network architecture**. However, as you rightly pointed out, their work remains **controversial** -- they were unable to surpass relpscost.
>
> > ### Subsequent Related Works -- From the Perspective of Neural Networks and Deep Learning
>
> Many subsequent works have been built upon these pioneering works [1,2,3]. Most of these works have adopted the feature design established by their predecessors, leading to a series of new research directions in branching policies, including but not limited to:
>
> 1. **New network architectures** [4]
> 2. **Updates in training paradigms**: Addressing issues in RL-based approaches [5,6]
> 3. **Enhancement or accelerated collection of expert datasets**: Improving quality [7,8] or efficiency [9] of data acquisition
> 4. **Hardware-friendly branching policies**: CPU-based rather than GPU-based implementations [10,11,12]
>
> The examples we've listed likely represent just the tip of the iceberg, and these works have all been published at top-tier conferences in the **neural network community** (including, of course, NeurIPS). We wish to emphasize that all these approaches examine the problem from a **neural network perspective**, solving MILP problems through **deep learning methodologies**. Compared to the pioneering works, they **did not propose new feature designs based on novel MILP insights**.

---

> ### Author Response · Authors · 2025-08-05
>
> > ### Clarification of Contributions
>
> We would like to reiterate that, **compared to TreeGate and T-BranT, we have achieved meaningful progress**. While **TreeGate and T-BranT fail to surpass relpscost** in all scenarios, our method demonstrates superior performance on **complex and meaningful** instances where **relpscost does not hold a significant advantage**. This addresses your concern: _"Why not just use relpscost directly?"_ In particularly challenging instances where relpscost itself struggles, **Mamba-Branching may indeed be worth considering**.
>
> We fully acknowledge that, compared to the **pioneering works** in this field, our contribution is more incremental. Our work primarily introduces a **new neural network/method from the perspective of the deep learning and neural network community** to improve branching policies, without making groundbreaking advances from the MILP community’s standpoint. However, many recent papers in this area -- published at **top-tier machine learning/deep learning conferences** (including but certainly not limited to [4–12]) -- have **similarly** focused on **enhancing branching policies purely from a deep learning perspective**.
>
> We completely **respect your viewpoint**: NeurIPS is one of the premier conferences for deep learning, and given your **rigorous academic standards** and **responsibilities as a reviewer**, you believe that pioneering works are more suitable for NeurIPS. This is entirely reasonable -- academic discourse naturally invites diverse perspectives ("A thousand readers, a thousand Hamlets"). **While we stand by our position, we deeply admire your scholarly rigor**.
>
> As you rightly noted, _"MILP optimization is inherently more complex than some other domains."_ We believe our discussion is **not a reflection of harsh criticism** but rather **a productive and profound academic exchange aimed at advancing the field**.
>
> > ### Acknowledgments
>
> Finally, we sincerely appreciate the time and effort you have devoted to this discussion. Your insights are thoughtful and deeply informed, and we are grateful for the valuable feedback you have provided to improve our work. Thank you for your contribution to our paper!
>
> > ### Reference
>
> [1] Khalil, Elias, et al. "Learning to branch in mixed integer programming." _Proceedings of the AAAI conference on artificial intelligence_. Vol. 30. No. 1. 2016.
>
> [2] Gasse, Maxime, et al. "Exact combinatorial optimization with graph convolutional neural networks." _Advances in neural information processing systems_ 32 (2019).
>
> [3] Zarpellon, Giulia, et al. "Parameterizing branch-and-bound search trees to learn branching policies." _Proceedings of the aaai conference on artificial intelligence_. Vol. 35. No. 5. 2021.
>
> [4] Ye, Xinyu, et al. "On Designing General and Expressive Quantum Graph Neural Networks with Applications to MILP Instance Representation." _The Thirteenth International Conference on Learning Representations_ (2025).
>
> [5] Scavuzzo, Lara, et al. "Learning to branch with tree mdps." _Advances in neural information processing systems_ 35 (2022): 18514-18526.
>
> [6] Parsonson, Christopher WF, Alexandre Laterre, and Thomas D. Barrett. "Reinforcement learning for branch-and-bound optimisation using retrospective trajectories." _Proceedings of the AAAI Conference on Artificial Intelligence_. Vol. 37. No. 4. 2023.
>
> [7] Zhang, Changwen, et al. "Towards imitation learning to branch for mip: A hybrid reinforcement learning based sample augmentation approach." _The Twelfth International Conference on Learning Representations_. 2024.
>
> [8] Gupta, Prateek, et al. "Lookback for Learning to Branch." _Transactions on Machine Learning Research_.
>
> [9] Lin, Jiacheng, et al. "CAMBranch: Contrastive Learning with Augmented MILPs for Branching." _The Twelfth International Conference on Learning Representations_ (2024).
>
> [10] Gupta, Prateek, et al. "Hybrid models for learning to branch." _Advances in neural information processing systems_ 33 (2020): 18087-18097.
>
> [11] Kuang, Yufei, et al. "Rethinking branching on exact combinatorial optimization solver: The first deep symbolic discovery framework." _The Twelfth International Conference on Learning Representations_. (2024).
>
> [12] Kuang, Yufei, et al. "Towards general algorithm discovery for combinatorial optimization: Learning symbolic branching policy from bipartite graph." _Forty-first International Conference on Machine Learning_. 2024.

---

> > ### Comment · Reviewer_QH4H · 2025-08-05
> > **Response to the Authors**
> >
> > Thank you very much for your thoughtful and comprehensive response. I appreciate the effort you've put into clarifying your contributions and providing a set of references from top-tier venues.
> >
> > I would like to offer a few clarifications and reflections from my side as well.
> >
> > **On Homogeneous vs. Heterogeneous MILPs**
> >
> > Many of the papers you cited indeed show that neural branching policies can outperform classical heuristics like relpscost, particularly in homogeneous MILP settings. In such cases, neural methods are often meaningful, as they clearly demonstrate potential to replace or improve upon traditional policies.
> >
> > However, when it comes to heterogeneous MILP instances—where the model is expected to generalize to unseen and structurally diverse problems—the challenge is considerably harder, as you also acknowledged. In such cases, classical heuristics like relpscost are strong.
> >
> > From this perspective, TreeGate is indeed a pioneering work, as it attempted to tackle this heterogeneous generalization problem. That said, I personally believe it also inadvertently set a trend that may have encouraged follow-up works which, while well-intentioned, tend to offer relatively incremental contributions. In many of these, the learned policies do not significantly outperform relpscost, and definitely cannot replace the expert policy.
> >
> > Thus, the core of my concern remains: if the learned policy is not able to significantly surpass relpscost on heterogeneous settings, the justification for adopting a neural policy becomes weaker. It becomes harder to argue that these neural policies are genuinely promising replacements rather than research prototypes with limited practical benefit.
> >
> > **Request for Top-tier Examples on Heterogeneous MILPs**
> >
> > To help me better appreciate your contribution in context, may I kindly ask: are there any published works at top-tier venues (NeurIPS, ICML, ICLR) that explicitly demonstrate neural branching policies outperforming relpscost in heterogeneous settings? This would provide valuable perspective and help position your work more clearly within the field.
> >
> > I fully acknowledge the difficulty of this generalization problem—and I agree with your view that this is an important and challenging frontier. If your method can show consistent advantages over relpscost in these scenarios, that would indeed be meaningful.
> >
> > ---
> >
> > Thank you again for engaging so constructively. I genuinely appreciate the academic dialogue and your openness in discussing the scope and limitations of the work. These kinds of discussions are essential for the healthy progression of research.

---

> > > ### Author Response · Authors · 2025-08-06
> > >
> > > Thank you for your response! I sincerely appreciate the opportunity to engage in this discussion with you, as I believe such academic exchanges are immensely beneficial for both our work and the advancement of our field.
> > >
> > > To the best of my knowledge, no published results in top-tier conferences (NeurIPS, ICML, ICLR) have yet surpassed relpscost in **heterogeneous** settings. This is precisely why we emphasize that Mamba-Branching’s ability to outperform relpscost on complex instances already represents meaningful progress.
> > >
> > > Of course, we acknowledge that surpassing relpscost across all instances -- particularly on simple ones -- remains challenging. This is because relpscost only exhibits limitations on complex instances (primarily those with a large number of variables).
> > >
> > > However, we believe that progress in most fields is often incremental, and breakthroughs rarely happen overnight. We would like to explore how our work and this discussion might inspire further developments in the field:
> > >
> > > 1. As you pointed out: The design of generalizable and strong-branching-adaptive features may be crucial -- likely requiring collaborative efforts between the MILP and deep learning communities.
> > >
> > > 2. Reinforcement learning and sequential modeling: To date, reinforcement learning methods for branching policies have not surpassed imitation learning approaches, largely due to the challenges posed by the complexity of the B&B environment. However, reinforcement learning inherently does not rely on expert demonstrations and holds the potential to surpass expert performance. In the future, sequential modeling ideas could potentially be integrated with reinforcement learning, following paradigms like Decision Transformer [1], to address long-term credit assignment challenges in B&B. In this context, Mamba-Branching might serve as an effective foundational network architecture.
> > >
> > > I believe this discussion will not only benefit our current paper but also greatly inform my future research in this field.
> > > Thank you once again for your active participation -- we have truly benefited greatly from this discussion.
> > >
> > >
> > > [1] Chen, Lili, et al. "Decision transformer: Reinforcement learning via sequence modeling." _Advances in neural information processing systems_ 34 (2021): 15084-15097.

---

> > > ### Author Response · Authors · 2025-08-07
> > > **Some Additional Results from our Latest Experiments**
> > >
> > > We sincerely apologize for the interruption! We would like to share some new experimental results and continue the discussion with you.
> > >
> > > Following Reviewer 3WGQ's suggestion, we conducted a set of experiments on the **homogeneous** problem. The purpose is to demonstrate that the **TreeGate+relpscost paradigm is not the fundamental reason why Mamba-Branching cannot consistently outperform relpscost** (only surpassing it in complex problems). Instead, **the inherent complexity of heterogeneous scenarios is the root cause**. If we **reduce the experimental difficulty**, **surpassing relpscost on homogeneous problems is not impossible**.
> > >
> > > Please allow me to quote our response to Reviewer 3WGQ as follows:
> > >
> > >
> > > > #### Results on homogeneous problems
> > >
> > > Many reviewers have raised concerns that imitating relpscost may not surpass relpscost itself. Therefore, here we conduct experiments on **homogeneous** scenarios. Specifically, we use **Set Covering** for both training and testing instances, only **varying the instance parameters**. The results are as follows:
> > >
> > >
> > > | Method | Mamba-Branching | TreeGate | Relpscost |
> > > |--------|-----------------|----------|-----------|
> > > | Nodes  | 265.07      | 280.38   | 170.81    |
> > > | Time   | **9.64**        | 9.89     | 11.96     |
> > >
> > > Table 1:  Results of training and testing on **Set Covering**. Both Mamba-Branching and TreeGate were initialized with relpscost once. During training, the expert policy being imitated was **relpscost**, as the previous section has already demonstrated that **imitating Strong Branching leads to significantly worse training accuracy**.
> > >
> > > This result clearly demonstrates one key point: **imitating relpscost does not inherently prevent a branching policy from surpassing relpscost**. However, why does Mamba-Branching only exceed relpscost on complex problems in heterogeneous scenarios? We argue that this is fundamentally due to the **inherent complexity of heterogeneous settings** -- all neural networks face generalization challenges, whereas relpscost remains consistent across any problem. **The intrinsic difficulty of heterogeneous scenarios leads to Mamba-Branching surpassing relpscost only in complex instances**. However, if the problem difficulty is reduced (e.g., by switching to homogeneous settings), exceeding relpscost is not impossible.
> > >
> > > Therefore, we would like to emphasize: **Mamba-Branching can surpass relpscost in complex heterogeneous scenarios, representing an improvement over prior work**. However, **the concern that Mamba-Branching does not consistently outperform relpscost stems from our deliberate choice of a challenging setting** -- heterogeneous scenarios. **In homogeneous problems, Mamba-Branching does achieve superior performance over relpscost**.

---

> > > > ### Comment · Reviewer_QH4H · 2025-08-08
> > > > **Response to the authors**
> > > >
> > > > Thank you for your big efforts in conducting these additional results on set covering benchmark. It is a really interesting finding that the features by TreeGate can lead to a neural policy which can achieve good performance on homogeneous MIPs.
> > > >
> > > > Some follow-up questions could be:
> > > > (1) Compared with GCNN on homogeneous settings, will Mamba-Branching have superiority?
> > > > (2) The authors could also conduct experiments following Gasse's GCNN paper's setting. If Mamba-Branching could achieve good performance in easy/medium/hard homogeneous instances (even better than GCNN), then I believe it could be a strong paper.
> > > >
> > > > I appreciate that the authors could run the experiments so quickly during the rebuttal phase. I maintain my score. But I suggest the author improve the paper by adding the experiment results about homogeneous settings. Then, it should be a strong paper in the next round/venue.

---

> > > > > ### Author Response · Authors · 2025-08-08
> > > > >
> > > > > Thank you for your response! Indeed, through experiments on homogeneous scenarios, we aimed to demonstrate that the TreeGate+relpscost approach is not inherently incapable of surpassing relpscost, but rather that it cannot consistently outperform relpscost in complex heterogeneous scenarios. As for the comparison between Mamba-Branching, TreeGate, and GCNN on homogeneous problems, we believe this deviates from our original intention: **to achieve breakthroughs in heterogeneous scenarios**. Under the TreeGate network specifically designed for heterogeneous scenarios, surpassing GCNN -- which is tailored for homogeneous scenarios—poses significant challenges, just as GCNN performs poorly in heterogeneous settings.
> > > > >
> > > > > However, your suggestion is indeed very valid -- success in homogeneous scenarios would undoubtedly make the paper more comprehensive. Certainly, if under the GCNN network setup (replacing TreeGate in Mamba-Branching with GCNN and substituting relpscost experts with strong branching), the approach could outperform GCNN, that would indeed be a highly compelling result. However, this experiment diverges significantly from our current setup -- it would constitute an entirely new study. Given the page constraints at NeurIPS, our focus on heterogeneous scenarios alone has already filled the manuscript to capacity. Introducing an entirely new set of experiments on homogeneous scenarios would make the content overly extensive. Moreover, if we were to focus solely on homogeneous scenarios, our contributions in surpassing relpscost on complex heterogeneous instances would be overshadowed.
> > > > >
> > > > >
> > > > > Thank you for the discussion. I have expressed my gratitude to you multiple times for your active engagement, which has been immensely beneficial to us.

---

### Official Review · Reviewer_KMMV · 2025-07-05

**Clarity:** 3
**Significance:** 3
**Originality:** 3
**Rating:** 3
**Confidence:** 4

**Summary:**

The paper studies the popular problem of using machine learning for branching in MILP solving. Different from many other papers, this paper pays attention to the features of the branch-and-bound tree instead of individual node features. To leverage tree features, it uses the sequence of node expansion and models it with the Mamba architecture. Empirical results show that the method expands fewer nodes than other state-of-the-art methods.

**Questions:**

1. What role does Proposition 1 play in the methodology? It seems to be simply a statement of a property that you wish to have. I don’t see how you achieve such property in the methodology. I.e., how do you enforce such a margin of \delta? Contrastive learning is a method that can approximate such thing, but it doesn’t enforce it given a \delta.
2. Why is strong branching rarely applied in practical scenarios? Why does relpscost provide a more realistic expert representation? Can you provide reference or more explanation? Most existing literature seems to still rely on strong branching to collect expert demonstration

**Ethical Concerns:**

["NO or VERY MINOR ethics concerns only"]

**Final Justification:**

I stand with my evaluation to reject the paper. The new results and rebuttal partially address my concerns but need another round of revision and review to determine if the paper meets the bar for NeurIPS acceptance.

**Limitations:**

See weakness 1.

**Quality:**

2

**Strengths And Weaknesses:**

Strengths:

1. Tree features are important in Branch-and-Bound. The paper leverages Mamba as a new architecture to inform branching decisions with those features. This is a novel aspect of the paper.
2. Experiments are done on real-world instances from MIPLIB and CORAL and look promising.
3. Source code is provided.

Weaknesses:

1. Experiment results are reported only on 8+73‎ = 81 instances. I can’t tell if the results are statistically significant. Especially on real-world instances, the runtime and number of nodes often has very high variance. Simply reporting the average of nodes and PD gap is not enough. I understand that difficult real-worlds instances are not that many. One way that you could obtain results on more instances is by rotating the train and testing sets. (For example, divide the instances into 4 groups and train on all combinations of three instance groups like similar to cross validation.) Also, you could evaluate the results on distributional settings like most other papers have done. You can obtain a lot more instances in distributional settings: https://sites.google.com/usc.edu/distributional-miplib/
2. The first ever paper on learning to branch needs to be in your reference: Learning to Branch in Mixed Integer Programming. Khalil et al., AAAI 2016.

---

> ### Author Rebuttal · Authors · 2025-07-28
>
> First, we appreciate your recognition of our strengths, including **the innovation of our research perspective, the emphasis on the evolution process of the branch-and-bound tree, the practicality tested on real datasets, and the support for open source**.
> Now, please allow me to address your pointed-out weaknesses and questions one by one.
>
> > ### Weakness 1: Insufficient experimental scale and lack of statistical significance analysis.
>
> We sincerely appreciate your feedback! In fact, two prior works in the same field [1, 2] employed **8** and **66** test instances, respectively, whereas our evaluation already includes a larger number of test instances.
> Moreover, these works also reported results using the **geometric mean**: given the high variance from inherent variability across test instances, it mitigates outlier impacts better than the arithmetic mean.
>
> To further address your concerns, we would be delighted to provide **additional statistical results** to demonstrate the statistical properties of our method.
>
> > #### Additional statistical information
>
> > ##### Quartile Tables
>
> | Method          | 25%   | 50%    | 75%    |
> |-----------------|-------|--------|--------|
> | Mamba-Branching | 48.17 | 140.33 | 390.84 |
> | T-BranT         | 83.54 | 228.26 | 855.90 |
> | TreeGate        | 55.22 | 140.85 | 617.96 |
>
> Table 1: For the easy instances in the MILP-L dataset (57 instances, each with 5 random seeds), the quartile table for time. For example, Mamba-Branching achieves the time of **140.33 at the 50%, meaning that 50% of the instances have solving time within 140.33**. Meanwhile, the solving time contains difficulty information, and from this table, we can observe **the adaptability of the method to problems of varying difficulties**.
>
>
> | Method          | 25%   | 50%    | 75%    |
> |-----------------|-------|--------|--------|
> | Mamba-Branching | 536.0 | 2505.0 | 7827.0 |
> | T-BranT         | 645.0 | 3109.0 | 8122.0 |
> | TreeGate        | 911.0 | 2928.0 | 8069.0 |
>
> Table 2: For the easy instances in the MILP-L dataset (57 instances, each with 5 random seeds), the quartile table for Node counts. The meaning of this table is similar to that of Table 1, and it can also indicate the adaptability of the method to problems of varying difficulties.
>
> As shown, Mamba-Branching attains the smallest time and node count at the 25th, 50th, and 75th percentiles, indicating consistently superior performance across all difficulty ranges compared to the other two methods. This suggests that Mamba-Branching’s lower geometric mean is not due to exceptional performance on a few extreme instances but rather stems from robust and balanced improvements across the entire benchmark.
>
> The primal-dual integral, which is related to bound values (independent of problem difficulty) and thus does not reflect problem complexity, is not included in our analysis but presented later with other statistical metrics. Additionally, the MILP-S benchmark, with only 8 test instances (too few for statistically rigorous analysis), was excluded from our statistical evaluation.
>
> > #### Win/Tie/Loss Statistics Table
>
> |                 | TreeGate | Mamba-Branching | T-BranT |
> |-----------------|----------|-----------------|---------|
> | TreeGate        | -        | 24/1/32         | 30/2/25 |
> | Mamba-Branching | 32/1/24  | -               | 33/1/23 |
> | T-BranT         | 25/2/30  | 23/1/33         | -       |
>
> Table 3: For the easy instances in the MILP-L dataset (57 instances, each with 5 random seeds), the Win/Loss table for node counts. Since 57 instances are too many to display their results directly, we use the win/tie/loss format to **show the median performance (across 5 random seeds) of different methods on various instances** when compared pairwise.
>
> |                 | TreeGate | Mamba-Branching | T-BranT |
> |-----------------|----------|-----------------|---------|
> | TreeGate        | -        | 23/0/34         | 37/0/20 |
> | Mamba-Branching | 34/0/23  | -               | 44/0/13 |
> | T-BranT         | 20/0/37  | 13/0/44         | -       |
>
> Table 4: For the easy instances in the MILP-L dataset (57 instances, each with 5 random seeds), the Win/Loss table for time. The meaning of this table is consistent with that of Table 3, both aiming to present the performance comparison of methods on each instance within the limited space.
>
> |                 | relpscost | Mamba-Branching | T-BranT | TreeGate |
> |-----------------|-----------|-----------------|---------|----------|
> | relpscost       | -         | 7/0/9           | 10/0/6  | 8/0/8    |
> | Mamba-Branching | 9/0/7     | -               | 14/0/2  | 13/0/3   |
> | T-BranT         | 6/0/10    | 2/0/14          | -       | 10/0/6   |
> | TreeGate        | 8/0/8     | 3/0/13          | 6/0/10  | -        |
>
> Table 5: For the hard instances in the MILP-L dataset (16 instances, each with 5 random seeds), the Win/Loss table for primal-dual integral. In our response to **Question 1 for Reviewer X7WR**, we have also specifically presented the **results for each instance**.
>
> It can be observed that Mamba-Branching outperforms both TreeGate and T-BranT on both easy and hard instances, while also achieving superior performance over relpscost on difficult instances. This finding aligns with the original conclusions presented in this paper.
>
> > #### Friedman test with Conover post-hoc analysis
>
> 1. Friedman test results for nodes (57 easy instances in MILP-L):
> 	Test statistic=2.502, p-value=0.2862, indicating no statistically significant differences.
> 2. Friedman test results for time (57 easy instances in MILP-L):
> 	Test statistic=18.000, p-value=0.0001, indicating statistically significant differences. Conover post-hoc analysis can be performed, as shown in Table 6.
> 3. Friedman test results for primal-dual integral (16 hard instances in MILP-L):
> 	Test statistic=8.625, p-value = 0.0347, indicating statistically significant differences. Conover post-hoc analysis can be performed, as shown in Table 7.
>
> |                 | Mamba-Branching | T-BranT  | TreeGate |
> |-----------------|-----------------|----------|----------|
> | Mamba-Branching | 1.000000        | 0.000014 | 0.071287 |
> | T-BranT         | 0.000014        | 1.000000 | 0.007329 |
> | TreeGate        | 0.071287        | 0.007329 | 1.000000 |
>
> Table 6: Conover post-hoc results for time (57 easy instances in MILP-L). The content of the table is p-values; if **a p-value is < 0.05**, it can be considered that there is a **significant difference**.
>
> |                 | Mamba-Branching | T-BranT  | TreeGate | relpscost |
> |-----------------|-----------------|----------|----------|-----------|
> | Mamba-Branching | 1.000000        | 0.011501 | 0.007880 | 0.114340  |
> | T-BranT         | 0.011501        | 1.000000 | 0.884271 | 0.310984  |
> | TreeGate        | 0.007880        | 0.884271 | 1.000000 | 0.247731  |
> | relpscost       | 0.114340        | 0.310984 | 0.247731 | 1.000000  |
>
> Table 7: Conover post-hoc results for primal-dual integral (16 hard instances in MILP-L). The meaning of this table is consistent with that of Table 6.
>
> The **Friedman test with Conover post-hoc analysis** is a non-parametric statistical method for comparing multiple methods across multiple instances, especially when data may violate normality assumptions.
> In fact, this approach is **a similar significance test method to the Wilcoxon signed-rank tests suggested by Reviewer X7WR in Weakness 1.**  However, it is **more suitable for our scenario** involving more than two methods.
> Its procedure includes: (1) calculating instance-wise ranks; (2) conducting the Friedman test for overall significance, and if p<0.05; (3) applying Conover's test for detailed comparisons.
>
> On easy instances,  in terms of runtime, Mamba-Branching demonstrates statistically significant differences compared to T-BranT (p=0.000014), while showing marginally significant differences with TreeGate (p=0.071287).
> For hard instances, Mamba-Branching has statistically significant differences vs. both TreeGate (p=0.011501) and T-BranT (p=0.007880), but no significant difference vs. relpscost (p=0.114340) -- here, reference to earlier statistical results is needed. This is reasonable as its performance advantage over relpscost is smaller than over other methods.
>
> > ### Weakness 2: Lack of citation to "Learning to Branch in Mixed Integer Programming"
>
> Thank you for your reminder and suggestions! This paper is the origin of learning to branch, and we will list it as an important work in the related work section.
>
> > ### Question 1: Role of Proposition 1
>
> Regarding this aspect, due to word count constraints, we have provided a detailed explanation in our response to **Reviewer TcMy's Question 2: Proof of Proposition**.
>
> > ### Question 2: The rationale for choosing relpscost over strong branching as the expert
>
> This hypothesis is entirely derived from [1], where the original text states:
>
> _"Our expert branching scheme is SCIP default, relpscost, i.e., a reliability version of hybrid branching in which SB and PC scores are complemented with other ones reflecting the candidates’ role in the search; relpscost is a more realistic expert (nobody uses SB in practice), and the most suited in our context, given the emphasis we put on the search tree."_
>
> Regarding why **relpscost is adopted as the expert over Strong Branching (SB) in generalization scenarios**, as well as **relpscost’s advantages over SB in dataset collection**, further explanations can be found in the following sections:
> 1. **"Response to Q3" for Reviewer QH4H**
> 2. **"Response to Weakness 4: Rationale for Expert Policy Selection" for Reviewer 3WGQ**
>
> > ### Reference
>
> [1] Zarpellon, Giulia, et al. "Parameterizing branch-and-bound search trees to learn branching policies." Proceedings of the aaai conference on artificial intelligence. Vol. 35. No. 5. 2021.
>
> [2] Lin, Jiacheng, et al. "Learning to branch with Tree-aware Branching Transformers." Knowledge-Based Systems 252 (2022): 109455.

---

> > ### Comment · Reviewer_KMMV · 2025-08-06
> >
> > Thank you for the rebuttal. I have read the rebuttal and decided to maintain my score.

---

### Official Review · Reviewer_3WGQ · 2025-07-14

**Clarity:** 4
**Significance:** 2
**Originality:** 3
**Rating:** 3
**Confidence:** 4

**Summary:**

This paper studies how to learn heuristics for guiding MIP solvers in their branch and bound process. Specifically, and as most works in this field, it focuses on the variable selection heuristic. Contrarily to most previous work, this approach takes inspiration from sequence modeling to solve the control problem of variable selection. The contribution lies primarily in the contrastive loss for representation learning of the token embedding, and in the usage of the previous Mamba architecture to process long sequences.

**Questions:**

Please see questions along the strengths and weaknesses part.

**Ethical Concerns:**

["NO or VERY MINOR ethics concerns only"]

**Limitations:**

Yes (and mostly non-applicable).

**Paper Formatting Concerns:**

None.

**Quality:**

2

**Strengths And Weaknesses:**

Strength. The paper is very well written and pleasant to read. The idea is sound and well presented.

Both strength and weakness. One strong advantage of sequence modeling for the resolution of MDPs is that it ignores the fact that the system to control might not be fully observable (and hence be a POMDP). For the case of variable selection in B&B solvers, this question is particularly important as the state variables have no guarantee of defining a Markov process. So using sequence modeling is an important good choice and a reason for acceptance of this contribution. However, the paper has little discussion on this topic: does the representation of Zarpellon et al. lead to an MDP? If so, then why would a sequence model perform better than a state-based policy? Conversely, if not, does the tree/variable token embedding preserve enough information for decision making (and why/how)? The sequence of token embeddings seems relevant to retrieve such necessary information but this receives little discussion of evaluation. There is a lot to be said here and the paper somehow lacks in this regard.

Weakness. As many works in this field, the contribution relies on imitation learning of an already known good heuristic (pseudo-cost branching and its variants), and conjectures that the sequence model will be able to generalize across instances. Although I believe the sequence modeling aspect is a really good idea, this last (implicit) conjecture deserves better discussion in my opinion. It is not specific to this paper but I find it lacking in most works (including this one) and I believe such studies should bring more to the table than empirical evidence that imitation learning yields good policies. Similarly, one aspect that is overlooked is how this could be included in an optimization process to solve the optimal control problem (in a dynamic programming loop for instance, of in a policy gradient method) rather than doing supervised imitation.

Moderate strength. I believe the contrastive learning approach is an interesting contribution. Focussing on representation learning for such problems is often overlooked and this contribution brings the question to light. The empirical evaluation (in particular the ablation study) seems to indicate this part of the contribution is important. I claim no expertise in the contrastive loss itself but the discussion on the matter is interesting.

Both (moderate) strength and (moderate) weakness. Exploiting the Mamba architecture is very important for this contribution, as it requires to process fairly long sequences and hence strongly relies on the fact that Mamba's forward pass has a complexity that is linear in the number of tokens (contrarily to the quadratic complexity of a Transformer architecture). One could argue that Mamba is not a contribution of this very paper and hence the contribution is limited, but I believe stitching together good ideas is a practice that makes interesting papers and hence is a (moderate) argument for acceptance. However I was a bit disappointed by the resulting sequence lengths. The paper reports using a length of 24 tokens across all models. This seems like a really small number if the sequence is to represent a hidden state of the MDP and retrieve the Markov property, or at least enough information for efficient decision making. This makes me doubt the ability of the contribution to generalize to difficult instances and to larger problems. Why would 24 tokens keep enough memory to accurately represent the state of the B&B tree? The absence of perspective on this aspect makes the contribution less significant.

Weakness. The empirical evaluation uses instances of MIPLIB and CORAL, which is good. However most works in this field use the Ecole benchmark as a testing ground. I believe Ecole's instances have important flaws and strongly appreciate seeing instances from MIPLIB. However, for evaluation fairness, reporting results on Ecole might provide interesting insights and strengthen the paper. So I believe the empirical evaluation is too weak in that regard (and also for the reasons that follow). Similarly, I find it surprising that the GCNN approach  the contribution is compared to is only that of Gasse et al. (2019). There have been alternate works, including that of Zarpellon et al., or Scavuzzo et al. (policy gradient method for branching policies), or Parsonson et al. (retrospective trajectories) and probably a few others that would yield a fair and insightful comparison. On the same page, if my memory is correct, Gasse et al. used strong branching as the expert to imitate, and fine-tuned their approach for this expert. Here the paper uses reliability pseudo-cost branching which is a stronger (and more complicated) heuristic. To fully and fairly assess empirically the importance of the sequence model and its ability to generalize (across experts and across instances), one would expect a comparison on several imitation experts. This is particularly important since most works that have tried to tackle both imitation learning or RL for such problems seem to struggle when it comes to imitating strong branching (or another heuristic). I also found the fact that average empirical results (e.g. Tables 1 and 2) are reported without standard deviation or any other variability metric across instances. This does not convince me that the proposed approach would perform well on all instances (despites its merits). And finally, since the central part of the contribution is the sequence model, I believe the paper would be much stronger if the properties of the model received more attention in the evaluation (and in particular the token sequence length as mentioned earlier). In particular, it seems quite surprising to train the sequence model on the very small number of instances contained in MIPLIB. This comes in stark contrast with the amount of data required to train such models generally and makes me doubt the relevance of the training loop. More details on why this training is sound and robust would be very welcome.

Lastly, the provided code does not seem to work. There's a "from mamba_ssm import..." in network.py but no mamba_ssm.py file anywhere to be found.

Overall I hope this feedback will help the authors, regardless of the outcome for the paper. I found the idea interesting and believe this is a relevant research direction that deserves the investigation effort.

---

> ### Author Rebuttal · Authors · 2025-07-26
>
> First, thank you for pointing out our strengths, including **the clarity of the paper writing and the design of sequence modeling and contrastive learning**. Below, we will elaborate on the weaknesses you identified and the questions you raised.
>
> > ### Response to Weakness 1: MDP or POMDP?
>
> We appreciate your feedback! Due to NeurIPS' page limits, we prioritized discussing sequential modeling and contrastive learning, leaving no space for analyzing whether the environment is a fully observable MDP or POMDP.
>
> In fact, the environment is an **MDP**; however, the performance of **sequence-based policies is superior to state-based policies**, which has been verified in **Decision Transformer (DT)** [5]. DT is still based on MDP, but it achieves better results by using historical trajectories as input. More importantly, as pointed out in [1], the B&B environment has a severe **long-term credit assignment problem**, where the impact of a branching decision may lag until the state many steps later. The original DT paper also emphasizes that their sequence modeling method effectively mitigates the long-term credit assignment problem, which is completely consistent with our approach.
>
> >### Response to Weakness 2: Generalization Concerns and Optimization Process
>
> We sincerely appreciate your insightful comments! Indeed, both points raised are profound, and we would like to address them individually.
> > #### Generalization Concerns
>
> In our work, generalization means the policy can train on certain instances and deploy on **completely unseen, heterogeneous ones** -- a significant improvement over GCNNs as shown in our results.
>
> This generalization ability benefits from the **TreeGate feature design (Zarpellon et al.) and expert selection**. TreeGate features are instance-independent, emphasizing the tree search process and thus incorporating numerous pseudocost-related terms. Meanwhile, based on the feature design, more appropriate relpscost is used as the expert. In contrast, GCNNs' features contain substantial instance-specific information, leading to poor generalization.
>
> > #### Optimization Process vs. Supervised Imitation
>
> As highlighted in references [1,2], the RL training process -- a most typical optimization process -- in B&B environments faces several fundamental challenges: including sparse rewards, credit assignment difficulties, and high variance in policy gradients caused by long episodes. To the best of our knowledge, for branching policies, **supervised imitation learning remains stronger than other training paradigms** in the current state of research.
>
> > ### Response to Weakness 3: Concerns Regarding Token Length
>
> We appreciate your attention to this implementation detail. We would like to clarify that **24 does not represent the token length**. As clarified in Section 4.2 (Line 201) of our manuscript, T=24 represents the total number of branching steps considered, while the actual number of tokens processed by Mamba is calculated as: $\sum_{t=0}^T |\mathcal{C}_t| + T$.
>
> For example, satellites1-25 (in MILP-L) has a SCIP solving time of 952.04 seconds, which can be considered a medium-difficulty instance. For this particular case, we conducted experiments where the **token count** reached **11,624** for **t=0~24**, which is a considerably large number.
>
> You might argue that considering only the past T=24 historical states and actions is insufficient. However, we justify this choice based on the following two points:
>
> 1. For complex problems where $|\mathcal{C}_t|$ is significantly larger, the token count would grow dramatically. You raised a valid concern that T=24 may not fully demonstrate the applicability to complex problems. However, we specifically chose T=24 precisely because complex problems typically involve more candidate variables, and this value represents a carefully considered trade-off.
> 2. The current state is indeed included in the input sequence. With T=24, our model receives strictly more input information than TreeGate's architecture.
>
> Thus, compared to TreeGate, our method incorporates **more comprehensive input information**, and T=24 was selected as a **necessary trade-off** to manage memory consumption effectively.
>
> > ### Response to weakness 4: Ecole Instances, Baselines, Expert Differences, Statistical Data, and Training Volume
>
> We sincerely appreciate your valuable feedback. These comments can be categorized into four key aspects:
> 1. The use of ecole instances and comparisons with other baselines
> 2. Differences in expert for imitation learning
> 3. Insufficient statistical information in the results
> 4. Concerns regarding the training data volume
>
> We will address each of these points systematically in the following response.
>
> > #### Why do we not use Ecole-Generated Instances nor compare with other baselines?
>
> Previous methods[1,2,3] often use Ecole-generated instances for **homogeneous problems** -- generating small-scale instances (randomized parameters) for training/testing and larger-scale instances for transferability evaluation, with Strong Branching as the expert to train GCNNs. **Ecole** here mainly serves to **generate diverse instances of the same problem**.
>
> In contrast, our training instances for MILP-S come from **19 distinct problems**, with expert data collected via relpscost, and test problems are **entirely different from these 19**. This is because Zarpellon et al.'s method (TreeGate) ensures the ability to handle heterogeneous MILPs through network and feature design, which GCNNs fundamentally lack due to their problem-specific feature design.
>
> Thus, TreeGate and GCNN are applied in completely different scenarios: **TreeGate focuses on solving unknown heterogeneous problems**, while **GCNN targets the same type of problem**. Hence, using **Ecole-generated instances** as test instances would be **meaningless** for our evaluation.
>
> As clarified earlier,  the papers you cited [1,2,3] all use **GCNNs** as their network architecture and do not aim for generalization. Our network architecture, following TreeGate, is fundamentally different. Nevertheless, we did include results from [3], which performed notably poorly. Since [1,2] share the same architectural limitations (unless their network designs are modified), their results would likely be similarly inadequate, making **further comparisons uninformative** for evaluating our contributions.
>
> > ####  Rationale for Expert Selection
>
> We acknowledge that relpscost is indeed less accurate than strong branching. Consequently, using relpscost as the expert policy for GCNN would inevitably degrade its performance.
> However, we deliberately chose Relpscost for the following critical reasons:
> 1. **Computational Tractability in Data Collection**:  Our dataset construction requires solving training instances with Relpscost to collect (state-action) pairs. While strong branching is more precise, its prohibitive computational cost makes it impractical for large-scale data generation on complex instances. Relpscost provides a feasible trade-off between efficiency and solution quality.
> 2. **Feature-Expert Alignment for Generalization**:  To ensure TreeGate's generalization capability across arbitrary instances, the feature design incorporates numerous pseudo-cost-related components. These features inherently align better with relpscost's behavior, as relpscost is a pseudo-cost that leverages high-quality historical information from strong branching. Using strong branching labels would create a mismatch between the features and expert policy, severely degrading model performance.
>
> We have also provided detailed explanations in **"Response to Q3" for Reviewer QH4H.**
>
> > #### Why No Standard Deviation?
>
> Since the test instances consist of various types of problems, each with inherent differences in difficulty and the number of nodes, the variance is not meaningful for our evaluation. Meanwhile, we use the geometric mean instead of the arithmetic mean to reduce the impact of extreme values across different problem types.
>
> We have also incorporated additional statistical information in **"Response to Weakness 1" for Reviewer KMMV.**
>
> > #### Real Data Volume
>
> As addressed in previous responses, the actual dataset does not simply consist of the number of instances. Rather, each instance contains a large collection of state-action data pairs. This point has been elaborated in detail in Line 238 of our manuscript. Specifically, the total number of training data pairs is **153,167 for MILP-S** and **393,668 for MILP-L**.
>
>
> > ### Regarding the issue with the code not running
>
> "mamba_ssm" is a Python library, not a file written by us, and you can install it via pip install.
>
> > ### Reference
>
> [1] Scavuzzo, Lara, et al. "Learning to branch with tree mdps." _Advances in neural information processing systems_ 35 (2022): 18514-18526.
>
> [2] Parsonson, Christopher WF, Alexandre Laterre, and Thomas D. Barrett. "Reinforcement learning for branch-and-bound optimisation using retrospective trajectories." _Proceedings of the AAAI Conference on Artificial Intelligence_. Vol. 37. No. 4. 2023.
>
> [3] Gasse, Maxime, et al. "Exact combinatorial optimization with graph convolutional neural networks." _Advances in neural information processing systems_ 32 (2019).
>
> [4] Chen, Lili, et al. "Decision transformer: Reinforcement learning via sequence modeling." _Advances in neural information processing systems_ 34 (2021): 15084-15097.

---

> > ### Comment · Reviewer_3WGQ · 2025-08-05
> >
> > Thank you for your feedback, and for the responses to the other reviewers' points, which I read with great interest.
> >
> > I do appreciate the effort you put into these answers, and still maintain my opinion on the strengths of the paper. Nonetheless, my main concerns remain. Useful representation learning of the B&B state remains an open question, and whether one can hope to obtain optimal stationary policies based on the features and architecture used in this work remains a conjecture (despite my interest for the ideas presented). Similarly, and as pointed out by other reviewers, I am not convinced imitating relpscost branching is actually a good way of demonstrating the effectiveness of mamba branching. Imitating strong branching on a set of smaller instances, and for homogeneous problems, would make for a way more convincing stepping stone, illustrating the ability to capture relevant features of the B&B tree development process, which would lift ambiguity as to why the presented approach performs well on larger scale and heterogeneous problems. Finally, I might still be missing something, but a time horizon 24 branching decisions makes for a very poor representation of the history of a B&B tree and this seems to defeat one of the core arguments for the presented (nice) idea.
> >
> > For these reasons, I will maintain my current appreciation, while encouraging the authors to keep on pushing this nice endeavor forward.

---

> ### Author Response · Authors · 2025-08-07
> **Response to Q1: Expectations on Useful Representation Learning**
>
> Thank you for your response! We sincerely appreciate your engagement in this discussion, and please allow me to address your concerns.
>
> > ### Response to Q1: Expectations on Useful Representation Learning
>
> We emphasize that the design of a branching policy consists of two key components: feature design and network architecture design. As extensively discussed with Reviewer QH4H, **feature design benefits from the MILP community**, while **network architecture design draws from advances in the deep learning community**.
>
> In our work, the feature design is entirely adopted from TreeGate, whereas our primary contribution lies in the neural network architecture: **a novel design based on Mamba for sequential modeling and contrastive learning.**
>
> Compared to learning representations directly from raw data (as in typical representation learning methods), the branch-and-bound tree does not possess readily available raw data. Instead, we must manually design a set of features to serve as the neural network's primary input. Effective features require deep domain knowledge of MILP solvers and careful, manual engineering.

---

> ### Author Response · Authors · 2025-08-07
> **Response to Q2: Reason for using Relpscost and Results on homogeneous problems**
>
> > ### Response to Q2: Reason for using Relpscost and Results on homogeneous problems
>
> Thank you for your suggestion! First, we would like to emphasize that our method focuses on **heterogeneous** scenarios (the premise), and TreeGate features is better suited for the relpscost expert. Nevertheless, we would be happy to conduct experiments in homogeneous settings as you suggested. We will address your concerns from these two perspectives separately.
>
> > #### Strong Branching or Relpscost? Further Comparative Experiments
>
> To **maintain the generalization capability of the branching policy** (this is a fundamental premise of our work, as we aim to focus specifically on **generalization** performance), we adopted the **same feature design and network architecture as TreeGate [1]**. Under this framework, we trained policies using both **strong branching** and **relpscost** as experts, with results below:
>
> |                             | Train Loss | Test Loss | Train Accuracy | Test Accuracy | Solving Time |
> | --------------------------- | ---------- | --------- | -------------- | ------------- | ------------ |
> | TreeGate + Relpscost        | 0.9954     | 0.9907    | 0.7453         | 0.7561        | 137.41       |
> | TreeGate + Strong Branching | 2.1845     | 2.5912    | 0.3590         | 0.2465        | 512.32       |
>
> Table 1: The imitation learning results using **TreeGate** as the network framework, with **strong branching** and **relpscost** serving as the experts respectively. The training was conducted for 50 epochs in all cases. The table reports the final network's loss and accuracy on both the training and test sets, as well as the solving time performance on test instances from MILP-S.
>
> The results demonstrate that under the TreeGate framework, **imitating strong branching achieves significantly lower accuracy compared to imitating relpscost**, making training considerably more difficult. Meanwhile, on test instances, the **policy imitating relpscost shows clearly superior performance**.
>
> > #### Results on homogeneous problems
>
> Many reviewers have raised concerns that imitating relpscost may not surpass relpscost itself. Therefore, here we conduct experiments on **homogeneous** scenarios. Specifically, we use **Set Covering** for both training and testing instances, only **varying the instance parameters**. The results are as follows:
>
>
> | Method | Mamba-Branching | TreeGate | Relpscost |
> |--------|-----------------|----------|-----------|
> | Nodes  | 265.07      | 280.38   | 170.81    |
> | Time   | **9.64**        | 9.89     | 11.96     |
>
> Table 1:  Results of training and testing on **Set Covering**. Both Mamba-Branching and TreeGate were initialized with relpscost once. During training, the expert policy being imitated was **relpscost**, as the previous section has already demonstrated that **imitating Strong Branching leads to significantly worse training accuracy**.
>
> This result clearly demonstrates one key point: **imitating relpscost does not inherently prevent a branching policy from surpassing relpscost**. However, why does Mamba-Branching only exceed relpscost on complex problems in heterogeneous scenarios? We argue that this is fundamentally due to the **inherent complexity of heterogeneous settings** -- all neural networks face generalization challenges, whereas relpscost remains consistent across any problem. **The intrinsic difficulty of heterogeneous scenarios leads to Mamba-Branching surpassing relpscost only in complex instances**. However, if the problem difficulty is reduced (e.g., by switching to homogeneous settings), exceeding relpscost is not impossible.
>
> Therefore, we would like to emphasize: **Mamba-Branching can surpass relpscost in complex heterogeneous scenarios, representing an improvement over prior work**. However, **the concern that Mamba-Branching does not consistently outperform relpscost stems from our deliberate choice of a challenging setting** -- heterogeneous scenarios. **In homogeneous problems, Mamba-Branching does achieve superior performance over relpscost**.

---

> ### Author Response · Authors · 2025-08-07
> **Response to Q3: Time Horizon**
>
> > ### Response to Q3: Time Horizon
>
> We understand your concerns, but the selection of 24 time horizons is a carefully considered **trade-off** given hardware constraints. We justify this choice from three perspectives:
>
> 1. **Hardware Efficiency**: As we mentioned in our previous rebuttal, while the 24-time horizon remains within reasonable limits, it already incurs substantial hardware overhead. Based on the experimental results, **this choice achieves performance improvements while maintaining reasonable computational costs**.
>
> 2. **Advancement Over Prior Work**: Compared to TreeGate, which only considers the current time horizon, our ability to incorporate 24 time horizons represents meaningful progress.
>
> 3. **Alignment with Motivation**: While it may appear contradictory at first glance, we emphasize that utilizing the entire sequence is not necessarily optimal. For instance, in reinforcement learning methods for POMDPs, RNN-based approaches often process the hidden state from the previous steps and the current observation -- despite having access to longer histories, RNNs employ forgetting mechanisms rather than retaining complete historical information. Similarly, for branch-and-bound trees, 24 time horizons strike a balance: they capture relevant sequential information while avoiding excessive focus on overly lengthy histories.
>
> In summary, the 24-time horizon achieves **hardware feasibility** while demonstrating **meaningful improvements** over prior work. Meanwhile, although sequential information is indeed crucial (as per our motivation) -- similar to other sequence modeling approaches (e.g., RNNs) -- it is neither necessary nor optimal to remember the complete history. The incorporation of **forgetting mechanisms aligns with, rather than contradicts**, our original motivation.

---

### Author Response · Authors · 2025-08-09
**Summary**

> ### Summary

We sincerely appreciate all the reviewers for their time and valuable comments on our work! First, we would like to express our gratitude for the positive recognition of our strengths, including: the clarity of the paper writing, the well-motivated and innovative nature of the work -- focusing on the expansion of B\&B trees, and the design of Mamba-based sequence modeling with contrastive learning methods. On heterogeneous real-world instances, Mamba-Branching has demonstrated superior performance.

Regarding our work, the reviewers have raised several common concerns. To be candid, some of these issues represent widely discussed topics in the broader field rather than being specific critiques of our paper -- nevertheless, we are more than happy to engage in a discussion with the reviewers about these points. We have summarized both the questions and our responses below.

---

> ### Author Response · Authors · 2025-08-09
> **Additional Statistical Information**
>
> > ### Additional Statistical Information
>
> Reviewer 3WGQ, KMMV, X7WR and TcMy have expressed concerns regarding the number of test instances in our experiments. In fact, the number of test instances **already exceeds** that of previous works (we have 57 + 16 = 73, while TreeGate [1] has 8 and T-BranT [2] has 66), which already demonstrates stronger generalizability.
>
> However, we have provided **additional statistical information** to further demonstrate the advantages of our method.
>
>
> > #### Quartile Tables
>
> | Method          | 25%   | 50%    | 75%    |
> | --------------- | ----- | ------ | ------ |
> | Mamba-Branching | 48.17 | 140.33 | 390.84 |
> | T-BranT         | 83.54 | 228.26 | 855.90 |
> | TreeGate        | 55.22 | 140.85 | 617.96 |
>
> Table 1: For the easy instances in the MILP-L dataset (57 instances, each with 5 random seeds), the quartile table for time. For example, Mamba-Branching achieves the time of **140.33 at the 50%, meaning that 50% of the instances have solving time within 140.33**. Meanwhile, the solving time contains difficulty information, and from this table, we can observe **the adaptability of the method to problems of varying difficulties**.
>
> | Method          | 25%   | 50%    | 75%    |
> | --------------- | ----- | ------ | ------ |
> | Mamba-Branching | 536.0 | 2505.0 | 7827.0 |
> | T-BranT         | 645.0 | 3109.0 | 8122.0 |
> | TreeGate        | 911.0 | 2928.0 | 8069.0 |
>
> Table 2: For the easy instances in the MILP-L dataset (57 instances, each with 5 random seeds), the quartile table for Node counts. The meaning of this table is similar to that of Table 1.
>
> As shown, Mamba-Branching attains the smallest time and node count at the 25th, 50th, and 75th percentiles, indicating consistently superior performance across all difficulty ranges compared to the other two methods. This suggests that Mamba-Branching’s lower geometric mean is not due to exceptional performance on a few extreme instances but rather stems from **robust and balanced improvements** across the entire benchmark.
>
> > #### Win/Tie/Loss Statistics Table
>
> |                 | TreeGate | Mamba-Branching | T-BranT |
> | --------------- | -------- | --------------- | ------- |
> | TreeGate        | -        | 24/1/32         | 30/2/25 |
> | Mamba-Branching | 32/1/24  | -               | 33/1/23 |
> | T-BranT         | 25/2/30  | 23/1/33         | -       |
>
> Table 3: For the easy instances in the MILP-L dataset (57 instances, each with 5 random seeds), the Win/Loss table for node counts. Since 57 instances are too many to display their results directly, we use the win/tie/loss format to **show the median performance (across 5 random seeds) of different methods on various instances** when compared pairwise.
>
> |                 | TreeGate | Mamba-Branching | T-BranT |
> | --------------- | -------- | --------------- | ------- |
> | TreeGate        | -        | 23/0/34         | 37/0/20 |
> | Mamba-Branching | 34/0/23  | -               | 44/0/13 |
> | T-BranT         | 20/0/37  | 13/0/44         | -       |
>
> Table 4: For the easy instances in the MILP-L dataset (57 instances, each with 5 random seeds), the Win/Loss table for time. The meaning of this table is consistent with that of Table 3.
>
> |                 | relpscost | Mamba-Branching | T-BranT | TreeGate |
> | --------------- | --------- | --------------- | ------- | -------- |
> | relpscost       | -         | 7/0/9           | 10/0/6  | 8/0/8    |
> | Mamba-Branching | 9/0/7     | -               | 14/0/2  | 13/0/3   |
> | T-BranT         | 6/0/10    | 2/0/14          | -       | 10/0/6   |
> | TreeGate        | 8/0/8     | 3/0/13          | 6/0/10  | -        |
>
> Table 5: For the hard instances in the MILP-L dataset (16 instances, each with 5 random seeds), the Win/Loss table for primal-dual integral.
>
> It can be observed that Mamba-Branching outperforms both TreeGate and T-BranT on both easy and hard instances, while also achieving superior performance over relpscost on difficult instances. This finding aligns with the original conclusions presented in this paper.

---

> > ### Author Response · Authors · 2025-08-09
> >
> > > #### Friedman test with Conover post-hoc analysis
> >
> > 1. Friedman test results for nodes (57 easy instances in MILP-L):
> >     Test statistic=2.502, p-value=0.2862, indicating no statistically significant differences.
> > 2. Friedman test results for time (57 easy instances in MILP-L):
> >     Test statistic=18.000, p-value=0.0001, indicating statistically significant differences. Conover post-hoc analysis can be performed, as shown in Table 6.
> > 3. Friedman test results for primal-dual integral (16 hard instances in MILP-L):
> >     Test statistic=8.625, p-value = 0.0347, indicating statistically significant differences. Conover post-hoc analysis can be performed, as shown in Table 7.
> >
> > |                 | Mamba-Branching | T-BranT  | TreeGate |
> > | --------------- | --------------- | -------- | -------- |
> > | Mamba-Branching | 1.000000        | 0.000014 | 0.071287 |
> > | T-BranT         | 0.000014        | 1.000000 | 0.007329 |
> > | TreeGate        | 0.071287        | 0.007329 | 1.000000 |
> >
> > Table 6: Conover post-hoc results for time (57 easy instances in MILP-L). The content of the table is p-values; if **a p-value is < 0.05**, it can be considered that there is a **significant difference**.
> >
> > |                 | Mamba-Branching | T-BranT  | TreeGate | relpscost |
> > | --------------- | --------------- | -------- | -------- | --------- |
> > | Mamba-Branching | 1.000000        | 0.011501 | 0.007880 | 0.114340  |
> > | T-BranT         | 0.011501        | 1.000000 | 0.884271 | 0.310984  |
> > | TreeGate        | 0.007880        | 0.884271 | 1.000000 | 0.247731  |
> > | relpscost       | 0.114340        | 0.310984 | 0.247731 | 1.000000  |
> >
> > Table 7: Conover post-hoc results for primal-dual integral (16 hard instances in MILP-L). The meaning of this table is consistent with that of Table 6.
> >
> > The **Friedman test with Conover post-hoc analysis** is a non-parametric statistical method for comparing multiple methods across multiple instances, especially when data may violate normality assumptions. Its procedure includes: (1) calculating instance-wise ranks; (2) conducting the Friedman test for overall significance, and if p<0.05; (3) applying Conover's test for detailed comparisons.
> >
> > On easy instances,  in terms of runtime, Mamba-Branching demonstrates statistically significant differences compared to T-BranT (p=0.000014), while showing marginally significant differences with TreeGate (p=0.071287).
> > For hard instances, Mamba-Branching has statistically significant differences vs. both TreeGate (p=0.011501) and T-BranT (p=0.007880), but no significant difference vs. relpscost (p=0.114340) -- here, reference to earlier statistical results is needed. This is reasonable as its performance advantage over relpscost is smaller than over other methods.
> >
> >
> > > ### Reference
> >
> > [1] Zarpellon, Giulia, et al. "Parameterizing branch-and-bound search trees to learn branching policies." Proceedings of the aaai conference on artificial intelligence. Vol. 35. No. 5. 2021.
> >
> > [2] Lin, Jiacheng, et al. "Learning to branch with Tree-aware Branching Transformers." Knowledge-Based Systems 252 (2022): 109455.

---

> ### Author Response · Authors · 2025-08-09
> **Proof of Proposition 1 -- Motivation for the Contrastive Loss**
>
> > ### Proof of Proposition 1 -- Motivation for the Contrastive Loss
>
> Reviewers KMMV, X7WR, and TcMy have all raised questions regarding Proposition 1, requesting further clarification on its connection to the contrastive learning approach and its role in our methodology, as well as potentially requiring a formal proof. In response, we have provided explanations from two perspectives: first, an intuitive explanation from the **conceptual standpoint**; second, an attempt to provide an explanation using **mathematical formulations**.
>
> First, we make the following two assumptions, which will serve as the foundation for our subsequent analysis:
>
> 1. The feature embeddings of the optimal branch decision variable ($e_t^a$) can be distinguished from those of other branch variables ($e_t^i, \forall i\in \mathcal{C}_t, i\neq a$) in the vector space.
> 2. The feature embedding $e_t$ is informative for branch decision-making.
>
> > #### Intuitive Explanation
>
> Based on assumption 1, we establish an anchor point $e_t^m$ and employ contrastive learning to train the model such that: (1) the optimal branch variable's feature $e_t^a$ is pulled closer to $e_t^m$, while (2) other variables' features are pushed away from $e_t^m$. This approach amplifies the separation between optimal and non-optimal features, thereby ensuring sufficient **distinguishability** when these features are subsequently processed by the Mamba architecture.
>
> > #### Mathematical Formulations
>
> For each branching variable $i$, we define $F(i)$ as the branching score function (in this paper, the **relpscost** score) for variable $i$, such that the optimal branching variable satisfies $$a = \arg\max_{i} F(i).$$
>
> Assumption 2 posits that the embeddings can encode branching quality information, which we simplify to a linear relationship $$e_t^i = \alpha F(i) + \epsilon_i,$$where $\alpha \in \mathbb{R}^d$ is a variable independent vector representing the quality-sensitive direction in the embedding space. $\epsilon_i \in \mathbb{R}^d$​ is a noise term satisfying $\mathbb{E}(\epsilon_i)=0$ with small magnitude $|\epsilon_i|$.
>
> We use max pooling over the embeddings as the anchor point, defined as:
> $$e_t^m = [...,\max_i e_t^i(j),...]
> =[...,\max_i (F(i)\alpha(j)+\epsilon_i(j)),...].$$
> For $e_t^a$, we have:
> $$e_t^a=[...,\max_i F(i)\alpha(j) + \epsilon_a(j),...].$$
> Indeed, if we assume that all $\epsilon_i$ terms are sufficiently similar in magnitude, we can derive the approximation $e_t^a \approx e_t^m$. Therefore, we can reasonably establish that
> $$\forall i \neq a,\text{sim}(e_t^a, e_t^m) > \text{sim}(e_t^i, e_t^m).$$
> This formulation is mathematically **equivalent** to the expression presented in the original paper.
> $$\forall t, \exists \delta > 0 \text{ s.t. } \text{sim}(e_t^a, e_t^m) \geq \max_{i \neq a} \text{sim}(e_t^i, e_t^m) + \delta$$
> Certainly, we acknowledge that the proof involves certain assumptions and approximations, which may lack full mathematical rigor. Additionally, during the neural network training process under contrastive loss, we cannot strictly guarantee that $\delta>0$ holds completely.
> However, we believe these results effectively **reflect the underlying intuition** behind our design: to make the optimal variable's features closer to the anchor point while pushing other features farther away, thereby making the optimal variable's features more **distinctive** in the vector space.

---

> ### Author Response · Authors · 2025-08-09
> **Rationale for Expert Selection: Strong Branching or Relpscost?**
>
> > ### Rationale for Expert Selection: Strong Branching or Relpscost?
>
> We note that Reviewers 3WGQ, KMMV, and QH4H have all raised concerns about our **use of relpscost as the expert** for imitation learning, with Reviewer QH4H particularly rejecting this approach entirely and suggesting that we should instead adopt the GCNN + strong branching expert training paradigm [3].
>
> We fully acknowledge that GCNN + strong branching represents a more commonly used and performance-proven approach. However, the application scenario of our work **differs fundamentally** from that of GCNN: while **GCNN focuses on training and testing on homogeneous instances**, our method specifically targets **testing on completely heterogeneous instances**, emphasizing the **generalization capability** of the branching policy.
>
> Therefore, we emphasize that our approach fundamentally relies on the TreeGate neural network architecture and feature design specifically tailored for **generalized branching policies**. Under this framework, we have experimentally validated **the necessity of using relpscost as the expert policy** and provided thorough **analysis of the underlying rationale**.
>
> We would like to **clarify** upfront that TreeGate+relpscost is **not our original contribution**, but rather represents the **standard paradigm** commonly adopted in prior work on generalized branching policies [1,2].
>
>
>
> > #### Comparative Experiments
>
> Under TreeGate, we trained policies using both **strong branching** and **relpscost** as experts, with results below:
>
> |                             | Train Loss | Test Loss | Train Accuracy | Test Accuracy | Solving Time |
> | --------------------------- | ---------- | --------- | -------------- | ------------- | ------------ |
> | TreeGate + Relpscost        | 0.9954     | 0.9907    | 0.7453         | 0.7561        | 137.41       |
> | TreeGate + Strong Branching | 2.1845     | 2.5912    | 0.3590         | 0.2465        | 512.32       |
>
> Table 1: The imitation learning results using **TreeGate** as the network framework, with **strong branching** and **relpscost** serving as the experts respectively. The training was conducted for 50 epochs in all cases. The table reports the final network's loss and accuracy on both the training and test sets, as well as the solving time performance on test instances from MILP-S.
>
> The results demonstrate that under the TreeGate framework, **imitating strong branching achieves significantly lower accuracy compared to imitating relpscost**. Meanwhile, on test instances, the **policy imitating relpscost shows clearly superior performance**.
>
> > #### Analysis and Clarification
>
> TreeGate emphasize incorporating **tree-search-related features** into the design without focusing on specific instances, while **pseudo-costs align with the requirements of tree search** by leveraging historical information. And relpscost essentially represents an improved form of pseudo-cost that incorporates historical strong branching experience.
>
> The features capture **"imperfect" pseudo-costs** while the relpscost labels contain **"refined" pseudo-costs**. The neural network learns to transform pseudo-costs **from imperfect to refined** versions. The feature design **inherently** results in better **compatibility** between TreeGate and relpscost, which consequently leads to the failure of TreeGate + strong branching.
>
> Moreover, relpscost outperforms strong branching in terms of **dataset collection speed**. Although compared to the fundamental limitations of strong branching, this advantage might be relatively less significant.
>
> Finally, we would like to **clarify** the following points:
> 1. **Our paper’s primary contribution lies in Mamba-based sequential modeling and contrastive learning**, rather than in re-evaluating the feature design choices of prior methods. Under the current framework, the TreeGate + strong branching is unsuitable. **Why don't we modify the feature design to accommodate strong branching?** This is **indeed a highly insightful suggestion**. However, such modification would extend **far beyond the scope of our current paper and represents an entirely new research direction** -- one that is both promising and challenging in its own right.
> 2. These comparative experiments are intended to **analyze TreeGate’s behavior under different experts**, rather than to serve as an ablation study for our proposed method.
>
>
> > ### Reference
>
> [1] Zarpellon, Giulia, et al. "Parameterizing branch-and-bound search trees to learn branching policies." Proceedings of the aaai conference on artificial intelligence. Vol. 35. No. 5. 2021.
>
> [2] Lin, Jiacheng, et al. "Learning to branch with Tree-aware Branching Transformers." Knowledge-Based Systems 252 (2022): 109455.
>
> [3] Gasse, Maxime, et al. "Exact combinatorial optimization with graph convolutional neural networks." _Advances in neural information processing systems_ 32 (2019).

---

> ### Author Response · Authors · 2025-08-09
> **Is it truly impossible to surpass relpscost itself by imitating relpscost? -- Experiments in Homogeneous Scenarios and the Fundamental Challenges in Heterogeneous Scenarios**
>
> > ### Is it truly impossible to surpass relpscost itself by imitating relpscost? -- Experiments in Homogeneous Scenarios and the Fundamental Challenges in Heterogeneous Scenarios
>
> Many reviewers have raised concerns that imitating relpscost may not surpass relpscost itself. Therefore, here we conduct experiments on **homogeneous** scenarios. Specifically, we use **Set Covering** for both training and testing instances, only **varying the instance parameters**. The results are as follows:
>
>
> | Method | Mamba-Branching | TreeGate | Relpscost |
> |--------|-----------------|----------|-----------|
> | Nodes  | 265.07      | 280.38   | 170.81    |
> | Time   | **9.64**        | 9.89     | 11.96     |
>
> Table 1:  Results of training and testing on **Set Covering**. Both Mamba-Branching and TreeGate were initialized with relpscost once. During training, the expert policy being imitated was **relpscost**, as the previous section has already demonstrated that **imitating Strong Branching leads to significantly worse training accuracy**.
>
> This result clearly demonstrates one key point: **imitating relpscost does not inherently prevent a branching policy from surpassing relpscost**. However, why does Mamba-Branching only exceed relpscost on complex problems in heterogeneous scenarios? We argue that this is fundamentally due to the **inherent complexity of heterogeneous settings** -- all neural networks face generalization challenges, whereas relpscost remains consistent across any problem. **The intrinsic difficulty of heterogeneous scenarios leads to Mamba-Branching surpassing relpscost only in complex instances**. However, if the problem difficulty is reduced (e.g., by switching to homogeneous settings), exceeding relpscost is not impossible.
>
> Therefore, we would like to emphasize: **Mamba-Branching can surpass relpscost in complex heterogeneous scenarios, representing an improvement over prior work**. However, **the concern that Mamba-Branching does not consistently outperform relpscost stems from our deliberate choice of a challenging setting** -- heterogeneous scenarios. **In homogeneous problems, Mamba-Branching does achieve superior performance over relpscost**.
>
> > #### Further Suggestion: Comparison with GCNN in Homogeneous Scenarios
>
> Currently, since we have demonstrated in homogeneous scenarios that TreeGate + relpscost can outperform relpscost, the reviewers further suggest comparing with GCNN under the same setting while following GCNN's experimental setup. However, we emphasize that **this would again entirely deviate from our original goal** -- **achieving breakthroughs in heterogeneous scenarios**. If we retain TreeGate + relpscost, a method specifically designed for heterogeneous scenarios, it will undoubtedly be uncompetitive compared to GCNN in homogeneous scenarios. However, if we completely abandon the TreeGate network and switch to using the GCNN network with the same experimental setup as GCNN, it would become an entirely new research direction -- requiring changes to the entire methodology and network design. Moreover, such a shift would deviate from the original focus of our work, which is dedicated to heterogeneous scenario research.
>
>
> We acknowledge that such a comparison would make our paper more compelling. However, due to the page limit of NeurIPS, our study on heterogeneous scenarios already occupies the full capacity of the manuscript. Adding an entirely new set of experiments in homogeneous scenarios would make the paper overly lengthy.

---

> ### Author Response · Authors · 2025-08-09
> **Without Comparison to Other Baselines and Other Training Paradigms**
>
> > ### Without Comparison to Other Baselines
>
> Reviewer 3WGQ mentioned that we should compare with baselines [1, 2], and Reviewer TcMy suggested a comparison with [3]. These works all employ neural networks for branching decisions, which aligns with our research domain, and we will include them in the related work section in the subsequent version.
>
> However, we would like to emphasize that **these works are not directly comparable to ours**. This is because **they are all based on GCNN frameworks**. As we mentioned earlier, these works focus on **solving isomorphic problems repeatedly -- where the same type of problem can have varying parameters but retains an unchanged structure**. In contrast, our method emphasizes a **generalized** branching policy: it is **trained on certain problems and then tested on completely unknown and heterogeneous instances**. This **distinction** is crucial, as evidenced by our experimental results, which show that **GCNN performs very poorly on heterogeneous instances**.
>
> Therefore, since [1,2,3] are all GCNN-based, their network architectures are **inherently unsuitable for heterogeneous-instance scenarios**. We believe that the comparison with GCNN is already sufficient, and further comparisons with these GCNN-based methods would not meaningfully validate our contributions.
>
> > ### Without Comparison to Other Training Paradigms
>
> Both reviewers TcMy and 3WGQ pointed out that we should consider comparisons with paradigms beyond imitation learning, such as the reinforcement learning (RL) paradigm. Indeed, reinforcement learning methods do not rely on expert demonstrations and thus have the potential to surpass imitation learning approaches.
>
> However, as discussed in [1,2], the characteristics of the branch-and-bound environment pose significant challenges for policy training under the RL paradigm, including:
> 1. **Sparse reward problem**
> 2. **Long-episode problem**: the excessive number of steps in an episode leads to high variance in policy gradients.
> 3. **Long-term credit assignment problem**: a single branching decision may have delayed effects that persist over many steps.
>
> To the best of our knowledge, **imitation learning remains the most effective paradigm** for training branching policies to date.
>
> > ### Reference
>
> [1] Scavuzzo, Lara, et al. "Learning to branch with tree mdps." _Advances in neural information processing systems_ 35 (2022): 18514-18526.
>
> [2] Parsonson, Christopher WF, Alexandre Laterre, and Thomas D. Barrett. "Reinforcement learning for branch-and-bound optimisation using retrospective trajectories." _Proceedings of the AAAI Conference on Artificial Intelligence_. Vol. 37. No. 4. 2023.
>
> [3] Seyfi, Mehdi, et al. "Exact combinatorial optimization with temporo-attentional graph neural networks." Joint European Conference on Machine Learning and Knowledge Discovery in Databases. Cham: Springer Nature Switzerland, 2023.

---

> ### Author Response · Authors · 2025-08-09
> **Additional Ablation Studies**
>
> > ### Additional Ablation Studies
>
> Reviewers X7WR and TcMy have each suggested additional ablation studies, including:
>
> 1. Replacing the contrastive learning with standard supervised learning, then discarding the linear classification head after training, named **Mamba-Branching-classifier**.
>
> 2. Adding **T-BranT-p**, which refers to T-BranT without the 1-step relpscost initialization, purely using the neural network.
>
> 3. Removing the positional encoding in Mamba-Branching, named **Mamba-Branching (w/o pe)**.
>
> 4. Replacing the sequence model with GRU, named **GRU-Branching**.
>
> Here, we systematically organize all supplementary results under both initialized and non-initialized (**-p**) settings for a comprehensive comparison.
>
> | Method     | Mamba-Branching | Mamba-Branching-classifier | TreeGate | Transformer-Branching | T-BranT |
> |------------|-----------------|----------------------------|----------|-----------------------|---------|
> | Nodes      | 2054.99         | 3353.60                    | 2171.31  | 3078.56               | 2668.62 |
> | Fair Nodes | 2077.55         | 3420.68                    | 2205.06  | 3120.04               | 2715.16 |
>
> Table 1: Performance comparison on MILP-S: All methods use one-time relpscost initialization. Mamba-Branching-classifier is added, which does not employ contrastive learning but instead uses standard supervised learning, with the linear classification head discarded after pre-training.
>
> | Method     | Mamba-Branching-p | Mamba-Branching-p-classifier | TreeGate-p | Transformer-Branching-p | T-BranT-p | GRU-Branching-p | Mamba-Branching-p (w/o pe) | Mamba-Branching-p (w/o cl) |
> |------------|-------------------|------------------------------|------------|-------------------------|-----------|-----------------|----------------------------|----------------------------|
> | Nodes      | 2272.43           | 4286.95    | 3179.55    | 5138.15                 | 3922.63   | 5683.86         | 3481.19                    | 3000.92                    |
> | Fair Nodes | 2272.43           | 4286.95    | 3179.55    | 5138.15                 | 3922.63   | 5683.86         | 3481.19                    | 3000.92                    |
>
> Table 2: Performance comparison on MILP-S: All methods rely purely on neural networks (-p), with TBranT-p, Mamba-Branching-p (w/o pe), and GRU-Branching-p added. Here, TBranT-p achieves pure neural network inference through algorithmic modifications, Mamba-Branching-p (w/o pe) removes positional encoding, and GRU-Branching-p replaces the sequence model with a GRU.
>
> The aforementioned results further demonstrate the advantages of Mamba-Branching, which is consistent with the conclusions drawn in the original paper.

---

### Note · Authors · 2025-08-11

> ### The Confusion in Evaluation Standards

Heterogeneous scenarios pose a challenging frontier: achieving generalization across fully diverse instances rather than similar problems. This is difficult but compelling.

Our decision to avoid homogeneous settings stems from experience. In past homogeneous work, we were often asked: “Why not test on MIPLIB?” We had to explain that homogeneous work uses structurally similar training and test instances, while MIPLIB is highly diverse. The follow-up critique was: “You only used toy datasets like set covering. If it can’t generalize to MIPLIB, it lacks value.”

To preempt this, we directly tackled heterogeneous scenarios -- far more challenging. We were encouraged to see Mamba-Branching outperform relpscost on complex heterogeneous instances, resembling real-world applications, as no prior heterogeneous method achieved this.

Yet now, in heterogeneous settings, we hear the opposite critique: "Since homogeneous methods have consistently outperformed relpscost, the limited progress in heterogeneous environments diminishes your contribution (failure to consistently surpass relpscost)." Even though we stressed that success in complex heterogeneous instances is meaningful given the inherent difficulty, reviewers still suggested reverting to homogeneous work.

However, a single paper cannot rigorously address both, as they require different networks and experimental setups. If we shift to homogeneous work, we risk again being asked for MIPLIB generalization -- reflecting the divergence in evaluation standards.

In short, homogeneous and heterogeneous research have entirely different evaluation standards, yet are often conflated. ML-for-branching seems stuck in a paradox: heterogeneous work is faulted for absolute performance, homogeneous work for generalization. Neither gains full recognition.

> ### Appeal for Recognition

We hope the committee can embrace our exploratory efforts. We kindly request that when evaluating this work, consider placing greater emphasis on the uniqueness and practical significance of heterogeneous scenarios, rather than demanding uniformly consistent performance as in homogeneous scenarios. Under the heterogeneous evaluation framework, our method has achieved meaningful progress by surpassing previous approaches.

> ### Gratitude

I would like to express my sincere gratitude to all reviewers, as well as to the PCs, SACs, and ACs for their dedicated efforts.

---

### Decision · Program_Chairs · 2025-09-17

**Decision:**

Reject

**Comment:**

This paper focuses on the development of variable selection heuristics for the branch-and-bound resolution of heterogeneous Mixed Integer Linear Programming (MILP) instances. In contrast to previous learning-based branching approaches, this study emphasizes the branching sequence from the root node to an optimal solution node. It incorporates two key ideas: utilizing the Mamba architecture to capture the temporal dynamics of branch-and-bound iterations, and employing a contrastive learning framework to pretrain the node embedding layer. The practical significance of this Mamba-Branching (MB) heuristic is demonstrated through various MILP instances.

The paper has received five reviews, with four borderline rejections and one borderline acceptance. While all reviewers agree on the clarity of the paper and the originality of the Mamba-driven approach, they also raised concerns regarding the justification for using contrastive learning and the statistical significance of the results. For these reasons, I cannot recommend acceptance.

Nevertheless, the rebuttal phase has involved extensive discussions between the authors and the reviewers. Notably, the additional justifications for contrastive learning, along with further experimentation highlighting the advantages of MB, are quite convincing and should be included in a revised version of this paper for future publication.